

# Greenhouse gas measurements from a UK network of tall towers: technical description and first results

Kieran M. Stanley[1*], Aoife Grant[1], Simon O'Doherty[1], Dickon Young[1], Alistair J. Manning[1,2], Ann R. Stavert[1], T. Gerard Spain[3], Peter K. Salameh[4], Christina M. Harth[4], Peter G. Simmonds[1], William T. Sturges[5], David E. Oram[5], Richard G. Derwent[6]

[1]School of Chemistry, University of Bristol, Bristol, United Kingdom
[2]Met Office, Exeter, Devon, United Kingdom
[3]Department of Experimental Physics, National University of Ireland, Galway, Ireland
[4]Scripps Institution of Oceanography, University of California San Diego, La Jolla, California, USA
[5]School of Environmental Sciences, University of East Anglia, Norwich, United Kingdom
[6]rdscientific, Newbury, Berkshire, United Kingdom

*Correspondence to*: Kieran M. Stanley (k.m.stanley@bristol.ac.uk)

**Abstract.** A network of tall tower measurement stations was set up in 2012 across the United Kingdom to expand measurements made at the long-term background northern hemispheric site, Mace Head, Ireland. Reliable and precise in situ greenhouse gas (GHG) analysis systems were developed and deployed at three sites in the UK with automated custom-built instrumentation measuring a suite of GHGs. The UK Deriving Emissions linked to Climate Change (UK DECC) network uses tall (165 - 230 m) open lattice telecommunications towers, which provide a convenient platform for boundary layer trace gas sampling. In this paper we describe the automated measurement system and first results from the UK DECC network for $CO_2$, $CH_4$, $N_2O$, $SF_6$, CO and $H_2$.

$CO_2$ and $CH_4$ are measured at all of the UK DECC sites by cavity ring-down spectroscopy (CRDS) with multiple inlet heights at two of the three tall tower sites to assess for boundary layer stratification. The long-term 1-minute mean precisions ($1\sigma$) of CRDS measurements at background mole fractions for January 2012 to September 2015 is $< 0.03$ $\mu$mol mol$^{-1}$ for $CO_2$ and $< 0.2$ nmol mol$^{-1}$ for $CH_4$. $N_2O$ and $SF_6$ are measured at three of the sites, and CO and $H_2$ measurements are made at two of the sites, from a single inlet height using gas chromatography (GC) with an electron capture detector (ECD) or reduction gas analyser (RGA). Repeatability of individual injections ($1\sigma$) on GC and RGA instruments between





January 2012 and September 2015 for $CH_4$, $N_2O$, $SF_6$, CO and $H_2$ measurements made using GC-ECD or -RGA were $< 2.8$ nmol mol$^{-1}$, $< 0.4$ nmol mol$^{-1}$, $< 0.07$ pmol mol$^{-1}$, $< 2$ nmol mol$^{-1}$ and $< 3$ nmol mol$^{-1}$, respectively.

Instrumentation in the network is fully automated and includes sensors for measuring a variety of instrumental parameters such as flow, pressures, and sampling temperatures. Automated alerts are generated and emailed to site operators when instrumental parameters are not within defined set ranges. Automated instrument shutdowns occur for critical errors such as carrier gas flow rate deviations.

Results from the network give good spatial and temporal coverage of atmospheric mixing ratios within the UK since early 2012. Results also show that all measured GHGs are increasing in mole fraction over the selected reporting period and, except for $SF_6$, exhibit a seasonal trend. $CO_2$ and $CH_4$ also show strong diurnal cycles, with night-time maxima and daytime minima in mole fractions. These data are used to produce independent emission estimates for comparison with the UK national inventory based on independent methods.

# 1 Introduction

Carbon dioxide ($CO_2$), methane ($CH_4$), nitrous oxide ($N_2O$), sulfur hexafluoride ($SF_6$), and carbon monoxide (CO) are potent greenhouse gases (GHG), which have a significant influence on the earth's climate system (Stocker et al., 2013). $H_2$ is an important indirect GHG, due to its photochemical reaction with hydroxyl radicals (OH) in the troposphere reducing OH mole fractions, thus increasing the lifetime of $CH_4$ and affecting ozone production (Grant et al., 2010a; Grant et al., 2010b; Luan et al., 2016). Atmospheric mole fractions of $CO_2$, $CH_4$, $N_2O$, $SF_6$ and CO have all exceeded pre-industrial levels due to anthropogenic activities (Kirschke et al., 2013; Stocker et al., 2013; Le Quéré et al., 2015). The increased concern about rising GHG emissions has already caused many nations to regulate their emissions. Inversion modelling techniques using data from atmospheric measurements can be used to derive emissions (Manning et al., 2011) and verify the national GHG inventories created using bottom



up approaches; however, the accuracy of the inversion is limited by the number and distribution of measurement locations available.

Remote measurements of GHGs first started in the 1950s at the Mauna Loa Observatory, Hawaii, USA. Remote background locations were chosen as to avoid strong anthropogenic sources encountered at stations close to populated regions which made data interpretation more difficult at the time (Keeling et al., 1976; Popa et al., 2010). Other background stations followed in the decades after Mauna Loa was set up, such as at Baring Head, New Zealand in 1970 (Brailsford et al., 2012) and the Atmospheric Life Experiment (ALE; a predecessor to the current Advanced Global Atmospheric Gases Experiment; AGAGE) in 1978 (Prinn et al., 2000). Measurements from these background stations only constrained global or hemispheric scale fluxes and were not able to capture local to regional scales (Gloor et al., 2001). Tall tower measurements in conjunction with transport models were proposed as a means to constrain local to regional scale GHG fluxes (Tans, 1993). GHG measurements from tall towers began in the 1990s (Haszpra et al., 2001; Popa et al., 2010) and have been expanded in the 2000s as part of a number of national and international measurement campaigns (Vermeulen, 2007; Kozlova et al., 2008; Thompson et al., 2009; Popa et al., 2010). Measurements made from ground level at terrestrial sites often display complex atmospheric signals with source and sink interactions visible. Sampling from tall towers reduces the influence of these local effects (Gerbig et al., 2003; Gerbig et al., 2009).

For over 30 years, high-frequency measurements of GHGs have been made at Mace Head (MHD), a global background measurement station on the west coast of Ireland. MHD predominantly receives well-mixed air masses, which have travelled across the Northern Atlantic, in the prevailing south-westerly winds, providing a good mid-latitude Northern Hemisphere background signal. These in situ, high-frequency, high-precision measurements have been used to estimate emissions of GHGs from the UK using the Inversion Technique for Emission Modelling (InTEM) methodology (Manning et al., 2011). In 2011, the UK government funded the establishment and integration of three new tall tower measurements stations in the UK. The UK Deriving Emissions linked to Climate Change (UK DECC) network was established to monitor the atmospheric mole fractions of GHGs, improve the spatial and temporal distribution of measurements across the UK and improve GHG emission estimates for



comparison with the national inventory, see Manning et al. (2011) for more details. The new network became operational in 2012. Of the four atmospheric monitoring stations, two main stations (MHD and Tacolneston: TAC) measure a suite of ~ 50 GHGs and ozone depleting substances (ODS; Table 1), while the two other stations (Ridge Hill: RGL; and Angus: TTA) measure the key GHGs. $CO_2$, $CH_4$,

$N_2O$, $SF_6$, CO and $H_2$ are the main focus of this paper. $CO_2$ and $CH_4$ are measured at all stations at high frequency (~ 3 seconds, except MHD which samples every 20 minutes), whilst $N_2O$, $SF_6$, CO and $H_2$ are measured at a lower frequency (detailed in Sect. 3).

The main objective of this paper is to describe an automated, reliable and high-precision analysis system for routine unattended monitoring of atmospheric $CO_2$, $CH_4$, $N_2O$, $SF_6$, CO, and $H_2$ within the

UK. We focus on the technical details of the network, review the performance of and present first results from the network.

## 2 Site location

The location of the three tall tower UK DECC stations was designed to provide good spatial measurement coverage across the UK utilising open lattice tall towers. Good spatial coverage was

necessary to provide information on emissions from the UK's devolved administrative regions of Scotland, Wales, England and Northern Ireland. The network consists of four sites all measuring key GHGs (Table 1). Instruments at the Irish coastal site at MHD take whole air samples from 10 meters above ground level (m.a.g.l.), whilst the three UK sites sample from differing heights on tall telecommunications towers (45 – 222 m.a.g.l.). The site locations and descriptions are given in Fig. 1

and Table 2, respectively. Minor instrumental changes have occurred within the network lifetime; however, the described instrumentation at the sites is correct as of September 2015.

### 2.1 Mace Head (MHD)

The MHD atmospheric research station is one of only a few western European stations that for significant periods of time is representative of mid-latitude Northern Hemispheric background air and

provides an essential baseline input for the UK DECC network. At the station (Fig. 1), numerous



ambient air measurements are made as part of the AGAGE (Cunnold et al., 1997;  Prinn et al., 2000), Integrated Carbon Observation System (ICOS) (Vardag et al., 2014) and the Global Atmospheric Watch (GAW) networks. Prevailing winds from the west to southwest sector bring well mixed background Atlantic air to the site on average 51 % of the time.  Polluted European air masses, as well as tropical

5    maritime air masses, cross the site periodically. MHD is uniquely positioned to observe these different air masses.  Galway, the closest city, has a population of ~ 75,000 and lies 55 km to the east. The area immediately surrounding MHD is very sparsely populated, providing very low local anthropogenic emissions.  The area surrounding MHD is generally wet and boggy with areas of exposed rock (Dimmer et al., 2001). The sample inlet is located 90 m inland from the shoreline (5 meters above sea level;

10    m.a.s.l.) and samples air from 10 m.a.g.l.. $CH_4$ and $N_2O$ measurements started at MHD on 23 January 1987. CO and $H_2$ measurements were added on 17 February 1994, and $SF_6$ measurements were included on 15 November 2003. A fully synoptic weather station operated by Met Eireann is located ~ 300 m from shore at 21 m.a.s.l..

## 2.2 Ridge Hill (RGL)

RGL is a rural UK site located 30 km east from the border of England and Wales (Fig. 1). It is 16 km south-east of Hereford (population 55,800), and 30 km south-west of Worcester (population 98,800), in Herefordshire, UK (ONS, 2012). The land surrounding the tower is primarily used for agricultural purposes and there are 25 waste water treatment plants within a 40 km radius of the site, the majority of which are in the northeast to south-easterly wind sector (DEFRA, 2012). Air samples are taken from

inlet lines located at 45 and 90 m.a.g.l. from a tall open lattice telecommunications tower at 204 m.a.s.l.. $N_2O$ and $SF_6$ measurements started on 1 March 2012 and are measured from 90 m.a.g.l. only, whilst $CO_2$ and $CH_4$ are measured from both heights sequentially and started on 23 February 2012.

## 2.3 Tacolneston (TAC)

TAC is a rural UK site located towards the east coast of England (Fig. 1). It is 16 km south-west of

Norwich (population 200,000), and 28 km east of Thetford (population 20,000), in Norfolk, UK (ONS, 2012). Lines sample air at 54, 100, and 185 m.a.g.l. from a tall open lattice telecommunications tower at





56 m.a.s.l.. $CO_2$ and $CH_4$ measurements started on 26 July 2012 and are measured from all three heights sequentially, whilst all other GHGs and ODSs (Table 1) are measured from the 100 m.a.g.l. inlet only. $N_2O$, $SF_6$, CO and $H_2$ measurements started on 26 July 2012. Land surrounding the tower is primarily used for agriculture, which is dominated by arable farming. Out of a total farmed area of over 400,000

hectares, 79 % of this is used in arable farming (DEFRA, 2010). There are three landfill sites between 30 and 50 km from the site, the closest being 30 km to the east (NCC, 2013). There is also a poultry litter power station in Eye, 20 km south of the site (EPRL, 2013).

### 2.4 Angus (TTA)

TTA is a rural UK site located near the east coast of Scotland (Fig. 1). It is 10 km north of Dundee

(population 148,000; GRO, 2013). A single line samples air at 222 m.a.g.l. from the tall open lattice tower at 400 m.a.s.l., which measures $CO_2$ and $CH_4$. Land surrounding the tower is predominantly under agricultural use, primarily livestock farming due to its hilly terrain. A Picarro G2301 was installed on 29 May 2013 and all TTA data reported in this paper is from 29 May 2013 to 30 September 2015 only.

### 3 Instrumentation

GHG measurement systems were developed in 2011 and then deployed in 2012 to enable measurements of GHGs from telecommunication towers within the UK. The system designs are similar to sampling equipment already deployed at Mace Head (Prinn et al., 2000) and at other tall tower sites (Popa et al., 2010; Winderlich et al., 2010). The systems are designed to utilise easily obtainable parts, so that rapid

replacement is possible on component failure, thus minimising system downtime and data gaps. This section outlines the instrumental setup used within the UK DECC network to measure GHGs. Table 1 summarises the trace gas species measured at each of the sites and the instrumentation used. Fig. 2 shows a schematic diagram for TTA, RGL and TAC, whereas the MHD setup is outlined in Prinn et al. (2000).





### 3.1 Sample tubing

At all UK DECC sites, instrumentation is located at the base of the towers in a building or a modified shipping container. At RGL and TAC, air is sampled through ½" O.D. 'Synflex 1300' or 'Dekabon' tubing (Hose Tech Ltd., UK), whilst at TTA, it is sampled through 3/8" 'Synflex 3000' tubing. This

tubing is made from high-density polyethylene bonded to overlapped aluminium tape and has a total wall thickness of 1.57 mm (Andrews et al., 2014). The outer polyethylene coating makes it resistant to water ($H_2O$) vapour condensation on the inner aluminium core tube. Air at MHD is sampled through ¼" stainless steel tubing (304 stainless steel, ¼" O.D., 0.209" I.D., Supelco, Sigma-Aldrich, UK). For the number of inlets at each site, please refer to Sect. 2. Tubing is held in place using UV-resistant plastic

clips or cable ties and runs down vertical metal tubes on the tower. Horizontal sections of tubing at the base of the tower were kept to a minimum and low points were avoided to prevent the accumulation of $H_2O$. For each inlet at RGL and TAC, an inverted stainless steel cup covers the inlet, acting as a shield to prevent $H_2O$ entering the line. A monel mesh screen is inserted within the cup to help prevent large particles from entering the inlet lines. The mesh screen was removed from the inlet cups at RGL in

September 2013 as it was thought that $H_2O$ was accumulating on the mesh and then being sucked into the inlet lines. This effect of the mesh promoting $H_2O$ entering the inlet lines has not been observed at TAC and the mesh is still in place. Perspex $H_2O$ decanting bowls with coalescing filter (Norgren, model F74G-4GN-QP3) are fitted to each sample line at its lowest point at the base of the tower to ensure that liquid $H_2O$ does not enter the laboratory and instrumentation. These $H_2O$ traps are checked on a weekly

to monthly basis and emptied manually using a toggle valve at the base of the decanting bowl.
Once the sample lines enter the laboratory, whole air samples pass through an inline 40 µm filter (SS-8TF-40, Swagelok, UK) to trap larger particles and then a 7 µm filter (SS-4F-7, Swagelok, UK) on the branched secondary instrument lines, according to the details in Fig. 2. The filters are present to protect instruments and pumps from particles (Andrews et al., 2014). Filters were not installed on the tower

inlets to prevent blockages from ice and subsequence system downtime. Unless stated otherwise, tubing within the laboratories is ¼" O.D., 0.209" I.D. 304 stainless steel (Supelco, Sigma-Aldrich, UK).



The sample line setup at TTA was different to the two other UK sites (RGL and TAC) as this site had previously been managed by the University of Edinburgh before being transferred to the University of Bristol (UoB) from January 2013. Specific differences at TTA include the sample inlet does not have a protective cup covering it (the air sampling line is cut on a bias so $H_2O$ has a drip point); and the $H_2O$

trap at the base of the tower is stainless steel rather than Perspex.

### 3.2 Pumps

Each sample line has its own dedicated oil-less linear pump (DBM20-801, GAST Group LTD., UK; TTA: Capex L2, Charles Austin, UK), continuously flushing at a flow rate of ~ 20 L min$^{-1}$, located downstream of all sampling equipment. Flow is measured downstream of the sample line pump using

flow meters (VFB-68, 3-30 L min$^{-1}$, Dwyer, UK) and vented into the laboratory. The continuous flushing of the inlet lines results in residence times within the tubing of between 7 and 35 seconds, depending on sampling height, from air intake to the instrumentation. At MHD, air is flushed at a flow rate of 5 L min$^{-1}$, with instruments sampling at rates between 100 to 110 mL min$^{-1}$. At TTA the sample line pump has a flow rate of 8.5 L min$^{-1}$. Branched secondary lines subsample from the main samples

lines for all instruments (Fig. 2).

Cavity ring-down spectrometers (CRDS) subsample from the main sample lines, passing through the sample selection system and a Nafion dryer (described in Sect. 3.3). CRDS instrument pumps (MD1 pump, Vacuubrand GMBH + Co KG, UK) are located downstream of the analyser. This has the advantage of eliminating sample contamination from the pump, reducing the likelihood of a torn

diaphragm introducing laboratory air into the sample and improving the performance of Nafion dryers. The CRDS outlet valve pressure is monitored as a diagnostic for instrument pump failure. The CRDS instrument pump uses Polytetrafluoroethylene (PTFE)/Viton® (also known as FKM) diaphragms. CRDS MD1 instrument pumps are located in the ambient internal laboratory air to allow efficient cooling from the fitted heat-sink.

All gas chromatograph (GC) systems and reduction gas analysers (RGA) use a similar line pump setup (described above), housed within a custom-built GC instrument sample module (Fig. 2). A KNF pump



(N86 STE, KNF Neuberger UK Ltd, Oxfordshire, UK) is located downstream of the diaphragm pump, which subsamples at a flow rate of 6 L min$^{-1}$. A Circor backpressure regulator (GO model LB1-2A01DCE171, Boiswood Ltd, UK) is used to control the KNF pump output pressure, which is viewed using a pressure gauge (0-60 psi Wika, Cole Palmer Instrument Co, UK) and a flowmeter (0.2-4 L min$^{-1}$, VFB-65, Dwyer, UK). This design enables the supply pressure of the air and standard to be matched and prevent pressure artefacts on the columns. Pressure and flow into the sample selection system is monitored by electronic pressure transducers and flow meters (details in section 3.4), meaning that the GC sample module performance can be monitored remotely and failures can be easily diagnosed.

### 3.3 Cavity ring-down spectrometer

$CO_2$ and $CH_4$ measurements at RGL, TAC and TTA are made using G2301 (Picarro Inc., USA) CRDS analysers (Tremblay et al., 2004; Crosson, 2008). Custom-made sample selection systems made at the UoB are used to switch between air inlets, calibration and standard gases for CRDS analysis, in addition to drying the samples (Fig. 2). The sample selection system consists of a 10 port multi-position valve (EUTACSD10MWEPH, VICI Valco AG International, Switzerland) to direct samples through the Nafion permeation dryer and to the CRDS. All air inlets and calibration gases are plumbed into the multi-position valve for ease of use and sample selection. Automatic sampling is achieved by controlling the multi-position valve using Linux based software (GCWerks$^{TM}$, www.gcwerks.com). An inline 2 μm filter (SS-4F-2, Swagelok, UK) is in place between the outlet of the front end system and the inlet to the analyser (Fig. 2) to further remove any particles that may negatively affect the CRDS.

The CRDS systems measure the decay time of the pulse of laser light inside a 35 cm$^3$ cavity at two wavelengths for $^{12}C^{16}O_2$ (1651 nm), $^{12}CH_4$ (1603 nm) and $H_2^{16}O$ (1603 nm) (Winderlich et al., 2010). Mole fraction measurements of each gas are provided at a frequency of ~ 3 seconds. Sample flow, temperature (318 ± 0.004 K) and pressure (140 ± 0.05 Torr) are maintained at specific set points as the size and shape of the spectral lines are sensitive to both temperature and pressure. The analyser is thus



designed to control temperature to a few hundredths of a K from 10-35°C and sample pressure to 0.05 Torr.

$H_2O$ can damage system components and interfere with measurements of GHGs, even at low levels (Andrews et al., 2014). $H_2O$ influences mole fractions of GHGs measured via CRDS through a dilution

effect, whereby a difference of 100 µmol mol$^{-1}$ (or parts per million; ppm) $H_2O$ can cause a "dilution offset" of 0.04 µmol mol$^{-1}$ in $CO_2$ (Andrews et al., 2014). Also, $H_2O$ vapour differences between calibration gases and air samples can cause spectral issues within optical instruments. $H_2O$ causes these spectral artefacts through line broadening effects on the spectroscopic line shapes, specifically Lorentzian broadening and Dicke line narrowing (Chen et al., 2010; Rella et al., 2013). The extent of

these effects is dependent on the atmospheric mixing ratio of $H_2O$ (Chen et al., 2010). To minimise these effects, samples were dried to < 0.25 % using permeation Nafion dryers (MD-050-72S-1, Perma Pure, USA) housed within the sample selection systems. The effectiveness of permeation Nafion dryers is dependent on the $H_2O$ partial pressure gradient between the sample and counter purge flows (Andrews et al., 2014). Therefore, dried zero air is used as the counter purge at 20 psi, supplied by a

compressor (JUN-AIR, model 2000, Norgren, Denmark at TAC and RGL, or DK50 plus, Ekom, Slovak Republic at TTA) connected to a zero air generator (TOC-1250, Parker Balston, USA).

In addition to using permeation Nafion dryers, a data correction can also be applied to remove spectral effects caused by $H_2O$ for CRDS systems. All CRDSs in the network produce data corrected for the $H_2O$ effects, using the correction coefficients listed in Rella (2010). The correction applied is minimised

due to the removal of most $H_2O$ using the Nafion dryer. TTA air samples were not dried prior to measurement until a Nafion drying system was installed in September 2014.

The CRDS instruments at TAC and RGL were fully operational for > 98 % for the time-period reported here. The TTA CRDS was operational for > 93 %. The inlet at Angus was not shielded from rainwater by an inverted cup, as was the case at other sites, and access to the site was more difficult due to its

remote location. As a result, the $H_2O$ trap filled up more frequently resulting in more ambient air data being rejected.





### 3.4 Gas chromatograph-electron capture detector

$N_2O$ and $SF_6$ were measured using gas chromatography coupled with micro electron capture detectors (GC-ECD) at RGL and TAC with a similar instrumental set-up (specific setup outlined in Table 3). A simplified schematic diagram for the GC-ECD systems at RGL and TAC is shown in Fig. 2. The GC-ECD at MHD measured $N_2O$ using a different experimental setup, details of which can be found in (Prinn et al., 2000), alongside details of the GC-FID instrument which measures $CH_4$ at Mace Head.

The $SF_6$ and $N_2O$ analysis method used at RGL and TAC was similar to that described in detail in Ganesan et al. (2013). A multi-position valve (EUTACSD8MWEPH, VICI Valco AG International, Switzerland; V1 in Fig. 2) directs calibration gas or sample air through a Nafion dryer (MD-050-72S-1, Perma Pure, USA) with a dry zero air counter purge (as outlined in Sect. 3.3), a second valve (V2 in Fig. 2; 8-port, 2-position valve; EUDAC8UWEPH, VICI Valco AG International, Switzerland) and into an 8 mL sample loop. The loop is flushed at 40 mL min$^{-1}$ for 60 seconds at a fixed exhaust pressure (~ 20 psi; Fig. 3 Backflush), volumetrically flushing the loop ~ 5 times. This second valve (V2 in Fig. 2) is used to control the injection of sample gas onto the columns prior to analysis. The inlet pressure for ambient air and calibration standards are matched to ensure reproducible sample introduction and equal pressure decay time to ambient pressure. Flow through the loop is controlled by a 'RED-y' smart series mass flow controller (GSC-A4TA-BB22, Voeglin Instruments AG, Switzerland) and pressure in the loop is measured using an 'All Sensor' pressure sensor (100PSI-A-DO, All Sensors, BS-Rep GmbH, Germany). The loop is allowed to decay to ambient pressure before injection through to an 8-port, 2-position valve (V3 in Fig. 2; EUDAC8UWEPH, VICI Valco AG International, Switzerland), which controls the sequential injection onto the pre-, main- and post-columns using P-5 carrier gas (a mixture of 5 % $CH_4$ in 95 % Ar; Air Products, UK).

The pre- and main columns, 1.0 m and 2.0 m, respectively (3/16" O.D. stainless steel packed with 80/100 mesh, Porapak Q), are contained in the main GC oven (Agilent GC-7890 at RGL and GC-6890 at TAC) at 90 °C, the post-column (0.9 m of 1/8" O.D. stainless steel packed with molecular sieve 5Å, 45/60 mesh) is housed in a thermostatically controlled heated inlet port of the GC at 180 °C. Separation of $N_2O$ and $SF_6$ from air occurs on the pre- and main columns after which a "heart-cut" is taken from



the flow to remove the majority of $O_2$ from the sample (Fig. 3). The $O_2$ is vented by switching V3 (Fig. 3) to backflush the pre-column with P5. This prevents the build-up of $O_2$ which may result in increased column bleed into the ECD resulting in increased baseline noise and reduced ECD sensitivity. The remaining, $O_2$ minimised, sample flows from the main to post-column where $N_2O$ and $SF_6$ are separated

further. The post column reverses the elution order of $SF_6$ and $N_2O$, preferentially retaining $N_2O$ as the 5Å pore size is too small for the $SF_6$ to physically enter. $SF_6$ molecules travel around the bulk molecular size with its retention time determined by the mesh size (Moore et al., 2003). This order of separation prevents the larger $N_2O$ peak from tailing into the small $SF_6$ peak, improving sample reproducibility and precision. Detection occurs in the micro-ECD which is held at 350 °C. The micro-ECDs at TAC and

RGL measure at a rate of 10 and 20 Hz respectively.

The main difference between the method used in our systems and those described by Hall et al. (2011) is the use of P5 carrier gas instead of $CO_2$ doped $N_2$. The use of P5 carrier gas enables the omission of $CO_2$ doping. $N_2O$ co-elutes with $CO_2$ on the column combination used within the UK DECC network, saturating the MS 5Å and providing a constant doping effect, thus care is taken to make sure the $N_2O$

response if not affected by this. Purity of the P5 carrier gas has previously been an issue where certain cylinders were found to be contaminated with $SF_6$ in varying amounts (5-80 pmol mol$^{-1}$, or parts per trillion; ppt). Each cylinder is now individually analysed as a sample to check for contamination prior to use.

The three valves (Valco universally actuated, RS-232 communication, purged housing) are controlled

remotely using GCWerks, enabling automatic sampling (see Sect. 3.8).

### 3.5 Reduction gas analyser

CO and $H_2$ are measured at two sites, MHD and TAC, using a RGA (RGA3 (MHD) and Peak Performer 1 (TAC), Trace Analytical Inc., USA). Table 4 outlines RGA instrumental setup at TAC and

MHD. The sample selection system is integrated within the GC-ECD system (Sect. 3.4) (Grant et al., 2010a; Grant et al., 2010b). The GC-ECD front-end system at MHD and TAC have a 10-port, 2-



position valve (VICI Valco AG International, Switzerland) for V2 (Fig. 2), instead of an 8-port 2-position valve, as at RGL. This allows for a 1 mL RGA sample loop to be put in sequence before the ECD sample loop (Fig. 3 TAC). After samples have been dried using the Nafion Dryer (MD-050-72S-1, Perma Pure, USA), passed through the sample loops and decayed to ambient pressure, they are injected

onto two isothermal packed columns held at 105 °C: a 0.768 m pre-column (1/8" O.D. stainless steel packed with 60/80 mesh Unibeads 1S) is used to protect the 0.768 m main column (1/8" O.D. stainless steel packed with MS 5Å, 60/80 mesh) from contamination by gases which adsorb to the surface of the main column and the main column separates $H_2$ and CO (Grant et al., 2010a). After separation, gases are injected into the RGA for analysis using zero air plus (Air Products, UK) carrier gas, where the

samples pass over a heated bed of mercuric oxide before being quantitatively determined using UV photometry (Grant et al., 2010a; Grant et al., 2010b).

### 3.6 Medusa gas chromatograph-mass spectrometer

The Medusa is a custom-built pre-concentration unit coupled to a gas chromatograph-mass spectrometer (GC-MS, the entire system is hereafter referred to as a Medusa GC-MS), which measures a wide range

of GHGs and ODSs. The Medusa GC-MS system is used at both MHD and TAC to measure $SF_6$, amongst other compounds. A detailed description of the Medusa setup is presented in Miller et al. (2008) for TAC and Arnold et al. (2012) for the $NF_3$ conversion (MHD setup). Briefly, a 2 L whole air sample is collected by the Medusa pre-concentration unit from the same sample pump as the GC-ECD (outlined in Sect. 3.6; Fig. 2) wherein the sample is dried using two Nafion dryers (MD-050-72S-1,

Perma Pure, USA) before being sequentially passed through two adsorbent traps cooled to -165 °C using a Cryotiger (Brooks Automation, Massachusetts, USA). More abundant gases (e.g. $N_2$, $O_2$, $CO_2$ and $CH_4$) are removed using temperature programming of the traps, allowing the trace species of interest to be isolated on the second refocusing trap after thermal desorption from the first trap. Trace gas species adsorbed on the second trap are released by heating the trap to 100 °C and passed through

two columns (three columns for $NF_3$ method at MHD) temperature programmed between 40 and 200 °C (Agilent 6890 GC, Agilent Technologies, UK) using helium carrier gas (MHD: BIP grade, Air





Products, UK; TAC: 6.0 grade, BOC, UK), separating out trace species chromatographically. Analytes are then detected via a quadrupole mass selective detector (Agilent 5973, Agilent Technologies, UK) in Selected Ion Monitoring (SIM) mode to increase sensitivity.

### 3.7 Logging, control and ancillary equipment

All instruments within the UK DECC network are controlled by GCWerks, installed on a local site computer running Ubuntu 12.04 LTS. GCWerks automates all instrument parameters (valves, trap and column temperatures, MSD, etc.), regulates switching processes, controls calibration cycles, displays chromatograms, performs peak integration and gives graphical and tabulated displays of all results. The automation of all instrumental processes helps to reduce problems and data loss associated with

connection problems between independent sample modules to instruments.

GCWerks generates automated user-specified alarms when instrument parameter conditions are not met. These alarms can also initiate instrument shutdown when specified to prevent instrumental damage.

The local site computer is connected to CRDS and GC analysers via Ethernet and the sample selection

systems communicate through serial (RS232) connections. Each site has a broadband internet connection which is utilised for remote access and control, automated data backup and maintaining system time synchronisation for each computer using the network time protocol. Data from instrumentation and ancillary equipment is logged and archived at all sites at a frequency of 0.3-20 Hz.

Uninterruptible power systems (UPS) are used at MHD (SG5K-6K, Falcon Electric Inc., USA), RGL

and TAC (Sentinel Dual SDL8000, Aiello UPS Ltd., UK) to prevent power surges and temporary power outages affecting instrumentation. The UPS provides up to 20 minutes of power to instrumentation in the event of a power outage. Additionally, an onsite generator provides continuous backup power at MHD with the UPS providing power for long enough to enable a seamless transition of power from line to generator.





## 4 Sampling and calibration

### 4.1 Sampling sequence

Sampling sequences within the network varies between instruments. CRDS instruments within the network are continuously measuring, with RGL and TAC measuring each sampling height sequentially

for 30 and 20 minutes, respectively, to ensure each sampling height is measured within each hour. The CRDS at TTA measures continuously from the single 222 m inlet. To ensure a good stabilisation period when sampling between different heights, the first 2 minutes of data after the valve switches to a new sample intake is automatically flagged out. The air sampling sequence is interrupted to analyse a daily standard gas and a monthly calibration sequence, outlined in Sect. 4.2.1.

The GC-ECD, RGA and Medusa GC-MS all have a lower sampling frequency than CRDSs. Sampling frequencies within the network are 10 minutes for the GC-ECD at RGL and TAC, 20 minutes for GC-FID and GC-ECD at MHD, 10 and 20 minutes for the RGA at TAC and MHD respectively, and 65 minutes for the Medusa GC-MS. Measurements alternate between ambient air and calibration gas, as outlined in Sect. 4.2.2.

### 4.2 Calibration

To guarantee the reliability and stability of measurements, automated calibrations are carried out periodically. Two separate calibration schemes are used, one for the CRDS and another for all other instruments. All tubing used for calibrant gases are 1/16" O.D., 0.03" I.D. 304 stainless steel (Supelco, Sigma-Aldrich, UK) to minimise dead volumes and wasted gas.

### 4.2.1 Cavity ring-down spectrometer calibration

CRDS instruments have two types of calibration standards, a standard of approximately ambient mole fraction and a set of calibration standards that span from below ambient up to elevated mole fractions. High-pressure aluminium tanks (Luxfer Gas Cylinders, UK) are used rather than steel to ensure long-term stability of $CO_2$ in the calibration gases. Regulator components may also have effects on the

25 stability of calibrations gases (Winderlich et al., 2010). Within the UK DECC network, regulators (64-




2640KA411, Tescom Europe) with polychlorotrifluroethylene (PCTFE) seals were used to prevent gas permeation. Calibrant and standard gases used in the CRDS instruments at all sites were filled and calibrated at GasLab MPI-BGC Jena and are of natural composition. $CO_2$ is on the WMO-X2007 scale (Zhao and Tans, 2006) and $CH_4$ is reported on the WMO-X2004A scale (Dlugokencky et al., 2005).

The standard gas is measured once a day for 20 minutes to assess for linear instrumental drift and the suite of calibration gases with varying mole fractions are measured once a month in quintuplicate to assess for instrument non-linearity and non-linear drift. The first five minutes of standard and calibration data and the first entire suite of calibration runs are removed to compensate for variability caused by regulator and line flushing. Table 5 details the standard and calibrant $CO_2$ and $CH_4$ mole

fractions currently used at RGL, TAC, and TTA. Linear interpolation between each daily standard gas analysis is used to remove instrumental drift and is performed automatically by GCWerks (see Sect. 5.1). Instrument non-linearity is assessed on a monthly basis by viewing the curve of the calibration gases and adjusted accordingly if there is a significant change in the curve. Instrument nonlinearity is also reassessed after changes in instrumental soft- and hardware. A second order non-linear curve is fit

to the data and implemented in GCWerks manually (see Sect. 5.1).

Measurements of daily standard gas showed a long-term precision (standard deviation, $1\sigma$, of 1 minute means from January 2012 to September 2015) of CRDS instruments within the UK DECC network of < 0.03 $\mu$mol mol$^{-1}$, and < 0.2 nmol mol$^{-1}$ (or parts per billion; ppb) for $CO_2$ and $CH_4$, respectively. CRDS precision within the UK DECC network is within the WMO compatibility guidelines for $CO_2$ ($\pm$ 0.1

$\mu$mol mol$^{-1}$) and $CH_4$ ($\pm$ 2 nmol mol$^{-1}$) (WMO-GAW, 2014).

### 4.2.2 Gas chromatograph calibration

The MHD GC-ECD, GC-FID, RGA, and Medusa GC-MS instruments and the TAC Medusa GC-MS are calibrated using tertiary standards. Working standards (also known as quaternary standards) are used to calibrate the Medusa GC-MS systems within the network and the GC-ECD and RGA at TAC and

RGL. Tertiary and quaternary standards are prepared by compressing background ambient air into 34 L electropolished stainless steel cylinders (Essex Cryogenics, Missouri, USA) using a modified oil-free



compressor (SA-3, RIX California, USA). Tertiary standards are filled at La Jolla, California, USA and calibrated at Scripps Institution of Oceanography (SIO) against their primary calibration scales via secondary working standards before being sent to MHD or TAC. Tertiary standards are also re-calibrated on return from site to assess each standard for sample stability over it's working lifetime.

Quaternary standards are filled at MHD and are calibrated/re-calibrated against the SIO calibrated tertiary standards at MHD on the GC-ECD and RGA before and after use at the tall towers. Mole fractions within the tertiary and quaternary standards are close to ambient background air sample values, minimising possible sample matrix non-linearities.

The quaternary standards are used to bracket air measurements on the GC-ECD, GC-FID, RGA and

Medusa GC-MS. Tertiary standards are used to bracket air measurements on the MHD GC-ECD/FID/RGA.  In addition, for the Medusa GC-MS, tertiary standards are analysed weekly and are used to calibrate the quaternary standards over the course of their use in the field. Quaternary standards last for two months to two years, depending on which instrument they are being used to calibrate, and tertiary standards last approximately eight months to two years. Studies have shown that no significant

drift of species contained in these standards occur over this time period (Hall et al., 2007; 2011). Calibration scales vary depending on the gas species, with $N_2O$ on SIO-98, $SF_6$ on SIO-05, $H_2$ on MPI-2009, and CO on the CSIRO04 calibration scales.

Repeatability of 20 minute injections of tertiary/quaternary standards ($1\sigma$) between January 2012 and September 2015 was < 2.8 nmol mol$^{-1}$ for GC-FID measurements of $CH_4$ (only at MHD), < 0.4 nmol

20    mol$^{-1}$ for $N_2O$ (GC-ECD), < 0.07 pmol mol$^{-1}$ for GC-ECD measurements of $SF_6$ and < 0.34 pmol mol$^{-1}$ for Medusa GC-MS $SF_6$ measurements, < 2 nmol mol$^{-1}$ for CO (RGA) and < 3 nmol mol$^{-1}$ for $H_2$ (RGA). CO measurements are within the WMO compatibility guidelines of $\pm$ 2 nmol mol$^{-1}$.

Due to the non-linear response of the GC-ECD to $N_2O$, non-linearity testing was carried out periodically in the field. Non-linearity testing was undertaken using a high mole fraction reference gas

(20 μmol mol$^{-1}$ $N_2O$ and 1 nmol mol$^{-1}$ $SF_6$ gas mix, BOC, Surrey, UK), which was dynamically diluted with zero air (Zero Air Plus, Air Liquide, Cheshire, UK) to the range of atmospheric mole fractions ($N_2O$: 240 – 400 nmol mol$^{-1}$; $SF_6$: 6 – 14 pmol mol$^{-1}$) using a custom-made dynamic dilution unit made





up of two RED-y mass flow controllers (GSC-A3TA-BB21, 100 mL min$^{-1}$; GSC-A4TA-BB22, 200 mL min$^{-1}$; Vögtlin Instruments AG, Switzerland). Results were used to create a second order non-linearity curve and a correction was implemented in GCWerks (see Sect. 5.2).

## 5 Data processing

GCWerks$^{TM}$ is used to process all of the CRDS, RGA and GC data. Raw measurement data and ancillary parameters stored on the local site computers are processed on site in near real time (NRT) for calibration and $H_2O$ corrections. Processing of data on site has the added advantage of aiding troubleshooting of instruments for site technicians.

Raw and processed data are mirrored daily from the local site computer to data processing servers at the
UoB or at the University of East Anglia (UEA) for TAC GC-ECD/RGA and Medusa GC-MS. Post processed UEA data is also mirrored to the UoB servers for archiving. All raw and processed data (calibrated and $H_2O$ corrected; Sect. 5.1-5.2) are subjected to QA/QC (Sect. 5.3), ensuring comparison with physical instrument parameters, such as CRDS cavity temperature and pressure, or flow rates, to check for spurious data. Data corrections outlined in Sect. 5.1 – 5.2 are investigated and implemented
on the processing servers and then mirrored back to the sites.

### 5.1 Algorithms for calculating $CO_2$ and $CH_4$

Raw CRDS data and ancillary parameters are acquired by GCWerks and are stored in binary strip-charts, named by UTC date, time, sample type (air, std, cal, tank) and inlet port number. These strip-charts contain all relevant data from the CRDS, i.e. wet and dry mole fractions, $H_2O$ mole fractions,
cavity temperature and pressure. Metadata stored within log files in GCWerks describes each sample type (air, std, cal), inlet height, how much data is to be rejected and data averaging frequencies for each Valco valve port within the sample selection system. Data within the network is averaged over 1, 5, 20 and 60-minute intervals, with time stamps corresponding to the beginning of the measurement, and stored. The first 2 minutes of air and 5 minutes of standard/calibration data after the Valco selector




valve switches within the sample system are automatically rejected to allow for stabilisation time and the tubing to condition with the new sample.

A number of data filters are automatically applied to the CRDS data before means are calculated. Data filters include cavity pressure and temperature out of normal operating range, $H_2O$ level too high, cycle
time too slow and the standard deviation in sample values being too great. Parameterisation of filters is generic for the type of analyser but can be user defined within GCWerks. For each single data point, these filter parameter values are verified and the data point discarded from the final dataset if not.

CRDS measurements are then corrected for linear instrumental drift and instrumental response over a span of different mole fractions, referred to here as non-linearity. Linear instrumental drift, monitored
by repeated measurements of a calibrated standard gas measured daily for 20 minutes, is corrected for by ratio of a measurement to the linear interpolation between bracketing standard measurements, as outlined in Eq. (1). Instrumental non-linearity is assessed using a function of the sample/standard ratio, outlined in Eq. (2). A second order function can then be fitted to the data to provide a non-linearity correction in Eq. (1).

$$C_{samp} = \frac{\left(\frac{R_{samp}}{R_{std}} \cdot C_{std}\right)}{NonLin} \qquad (1)$$

$$NonLin = f\left(\frac{R_{cal}}{R_{std}}\right) = \frac{\left[\frac{R_{cal}}{C_{cal}}\right]}{\left[\frac{R_{std}}{C_{std}}\right]} \qquad (2)$$

where $C_{samp}$ is the calibrated $CO_2$ or $CH_4$ mole fraction, $R_{samp}$ and $R_{cal}$ are the sample and calibrant raw dry-air mole fraction from the CRDS, respectively, $R_{std}$ is the linear interpolation between the raw dry-air mole fraction of the two bracketing standards, $C_{std}$ and $C_{cal}$ are the calibrated mole fraction
assigned at GASLAB MPI-BGC Jena, and $NonLin$ is the non-linearity correction coefficient assigned by the user from the second order fit of calibration data.

### 5.2 Algorithms for calculating $N_2O$, $SF_6$, CO and $H_2$

GC data and ancillary parameters are acquired by GCWerks and stored in chromatograms and strip-charts, named by UTC date, time, sample type (air, std, cal, tank) and inlet port number. GC




chromatograms are acquired and displayed in real time, and are stored with a 4:1 compression ratio. Temperature (ambient and sample selection module), loop flow rates and pressures at the time of sample injection onto the columns are also stored in a sample log file with the corresponding date and time.

User defined integration parameters allow for automatic integration of peaks. Chromatograms can be reprocessed for selected periods when peak integration parameters need to be altered due to changes in baseline and retention times. Integrated peak heights and areas are stored and used along with pressure and temperature data stored in the sample log file to calculate mixing ratios.

$N_2O$, $SF_6$, CO and $H_2$ are calibrated for linear instrumental drift and non-linearity in a similar way to
measurements from the CRDS instrument. A variation of Eq. (1) is used to calibrate data for linear instrumental drift using sample integrated height or area (Eq. (3)). The non-linearity fit is defined using the dynamic dilution of a high concentration cylinder, as described in sect. 4.2.2 and implemented using Eq. (4).

$$C_{samp} = \frac{RL \cdot C_{std}}{NonLin} \qquad (3)$$

$$RL = \frac{R_{samp}}{R_{std}} \cdot \frac{\frac{P_{samp} \cdot V}{R \cdot T_{samp}}}{\frac{P_{std} \cdot V}{R \cdot T_{std}}} \qquad (4)$$

where $C_{samp}$ is the calibrated $N_2O$, $SF_6$, CO and $H_2$ mole fraction, $RL$ is the sample/standard ratio, $R_{samp}$ is the sample raw dry mole fraction, $R_{std}$ is the linear interpolation between the raw dry-air mole fraction of the two bracketing standards, $C_{std}$ is the calibrated standard mole fraction, $NonLin$ is the nonlinearity correction coefficient assigned by the user from the second order fit of calibration data,
$P_{samp}$ and $P_{std}$ are the sample and standard loop pressures, respectively, at the time of sample injection, $T_{samp}$ and $T_{std}$ are the sample and standard gas temperatures, respectively, at the time of injection, $V$ is the loop volume and $R$ is a gas constant.





### 5.3 Final data processing

In the first phase, chromatograms and strip-charts are reviewed daily, on a site-by-site basis to check for good integration and systematic biases not detected by automatic data processing routines. Filtered data can also be reviewed in strip-charts to ensure good filter parameterisation and ensure non-spurious data

isn't unnecessarily filtered out. Data and ancillary measurements are reviewed in parallel to help observe potential errors and diagnose issues within the data. Instrument precision is reviewed by monitoring the standard gas concentrations for anomalies. 1, 5, 20 and 60-minute mean (CRDS) or discrete (GC-ECD/FID/RGA and Medusa GC-MS) air data can also be plotted against instrumental and ancillary parameters to further investigate data issues. Spurious data are flagged and a justification for

the flagging of data is given and logged.

In the second phase, data from the entire network are reviewed simultaneously in GCcompare, custom built data visualisation software. Flagged GCWerks time series data from the network are overlain to compare sites with the background station (MHD) and to look for differences between sites. Potential issues not previously noted are investigated using ancillary and instrumental parameters, as well as air

history maps produced on an hourly basis using the Numerical Atmospheric dispersion Modelling Environment (NAME) Lagrangian dispersion model outlined in Manning et al. (2011).

### 6 Results

Measurements of GHGs from the UK DECC network are presented from January 2012 through to September 2015 (Fig. 4 to 6; see Sect. 2 for details of start dates of data acquisition). Results shown in

this paper are limited to qualitative analyses of the most prominent features of the data; utilisation of this large and comprehensive dataset to its full potential lies in the use of high resolution inverse atmospheric transport models (Manning et al., 2011; Vermeulen et al., 2011; Ganesan et al., 2015). All $CO_2$ and $CH_4$ data are publicly available as hourly means, whilst $N_2O$, $SF_6$, CO and $H_2$ are available as discrete samples, at EBAS, as database infrastructure operated by the Norwegian Institute for Air

Research (http://ebas.nilu.no/) and the World Data Centre for Greenhouse Gases




(http://ds.data.jma.go.jp/gmd/wdcgg/). All MHD data, except CO mole fractions, are available from the Carbon Dioxide Information Analysis Centre (CDIAC) at http://cdiac.ornl.gov/.

### 6.1 Seasonal cycles

$CO_2$ shows the most marked seasonal cycle of all the GHGs measured in the UK DECC network, due to its major biogenic uptake via photosynthesis and production from respiration, as well as anthropogenic sources. The approximate amplitude for mid-latitude northern hemisphere seasonal cycle is 15 µmol mol$^{-1}$ and has an upwards trend. Fig. 4(a) shows $CO_2$ maxima in January/February and minima in August. Sites show a $CO_2$ signal that varies in a "noise" band of approximately 20 µmol mol$^{-1}$ (~ 5 %), alongside a strong seasonal cycle. Large differences between the sites can also be observed from $CO_2$ data in Fig. 4(a). TTA shows the lowest frequency and magnitude of above baseline events. This is thought to be a combination of the tower inlet height, which is at 222 m.a.g.l. compared to lower inlets at RGL (45/90 m.a.g.l.) and TAC (54/100/185 m.a.g.l.), and the location of TTA in the north of the UK in a much more sparsely populated region than the other sites. TAC $CO_2$ mole fractions have greater excursions from baseline compared with the other sites due to its eastern location, downstream of the predominant south-westerly wind direction, and location near to a number of large sources.

$CH_4$ also shows seasonal variation, see Fig. 4(b), with a winter maxima and a summer minima, driven by greater oxidation by hydroxyl radicals in strong sunlight and greater uptake from the troposphere by methanogenic bacteria in soils (Dlugokencky et al., 2011). The approximate amplitude for mid-latitude northern hemisphere seasonal cycle is 21 nmol mol$^{-1}$ and has an upwards trend. Differences between site baseline (unpolluted) mole fractions can be observed in Fig. 4(b). The relative variability in $CH_4$ atmospheric signal within the UK DECC network varies roughly between 1800 – 2300 nmol mol$^{-1}$. MHD and TTA generally agree well over the observation period and have the lowest frequency and magnitude of pollution events. This is thought to be due to fewer pollution sources within the prevailing wind direction at both MHD and TTA. As per $CO_2$, TAC has the greatest excursions in $CH_4$ mole fraction compared with the other sites in the network due to its location downwind of major urban areas such as London and Birmingham in prevailing wind directions.



A seasonal cycle is also observed in $N_2O$ mole fractions, see Fig. 5(a); however, this seasonality is less well defined than for $CO_2$, especially in 2014. There is an approximate 0.8 nmol mol$^{-1}$ northern hemisphere mid-latitude seasonal trend. Summer minima in $N_2O$ are thought to be caused by the descent of stratospheric air bringing $N_2O$-depleted air into the troposphere across the polar tropopause

(Nevison et al., 2011). The atmospheric signal has an upward trend and varies between 322 – 338 nmol mol$^{-1}$. Like $CO_2$ and $CH_4$, magnitude and frequency of $N_2O$ pollution events are greater at RGL and TAC than MHD. This is thought to be due to the surrounding land being used predominately for agriculture and fertiliser application causing nitrification and denitrification to occur.

Tropospheric $SF_6$ mole fractions do not show a seasonal cycle but a clear increase over time, see Fig.

5(b). The atmospheric variability within the network is between 7.5 – 13 pmol mol$^{-1}$. The magnitude and frequency of $SF_6$ pollution events at RGL and TAC are greater than at MHD as emissions are predominantly from anthropogenic sources. Natural sources of $SF_6$ are considered to be so low they can be ignored (Levin et al., 2010).

A seasonal cycle is also observed in CO and $H_2$ at MHD and TAC (Fig. 6). There is an approximate 37

and 36 nmol mol$^{-1}$, for CO and $H_2$ respectively, northern hemisphere mid-latitude seasonal trend.  CO has winter maxima and summer minima, driven predominantly by anthropogenic emissions and the strength of the summertime OH sink (Grant et al., 2010b; Satar et al., 2016). $H_2$ has delayed spring maxima and autumn minima due to maximum summertime loss by OH oxidation and greatest rates of soil uptake in the summer and early autumn when soils are driest (Grant et al., 2010b).

**6.2 Diurnal cycles and vertical gradients**

In principal, measurements of GHGs at different heights at a station allow observations of sources and sinks from different spatial footprints (Vermeulen et al., 2011). The influence of local source of GHGs is greatest at night and the early morning, as observed in Fig. 7, with higher mixing ratios at lower inlet heights caused by the low night-time planetary boundary layer heights. The average diurnal profiles for

$CO_2$ and $CH_4$ at RGL and TAC (Fig. 7) are from 1 January 2012 to 30 September 2015. The summer data shows a strong diurnal cycle with $CO_2$ maxima at night-time when respiration (emissions)



dominates and $CO_2$ minima at mid-day when photosynthesis (uptake) dominates. Night-time $CO_2$ is greatest during the winter and spring months, with maxima around 06:00 UTC. Winter months have a smaller diurnal cycle than other seasons and a delayed uptake of $CO_2$ (approximately 09:00 UTC) in comparison to summertime uptake (06:00 UTC), caused by reduced rates of biosphere assimilation and

5 later sunrises.

The greatest difference in $CO_2$ mole fraction between the lowest and highest inlets were 3 and 8 μmol mol$^{-1}$ at RGL and TAC respectively. The highest mole fractions were observed at the lowest inlet heights, due to local net $CO_2$ emission in the area of the inlet footprint during night-time and a lower boundary layer heights. Daytime vertical differences were very small for all seasons (< 1 μmol mol$^{-1}$)

for both TAC and RGL. Spring, summer and autumn daytime concentrations are generally lower at the lowest inlet heights for RGL and TAC due to the net biospheric $CO_2$ uptake within the footprint area. $CO_2$ uptake during the daytime was also observed during winter months, although this is less pronounced than other seasons and wintertime average daytime $CO_2$ concentrations are always greater at the lowest inlet height. These seasonal patterns in vertical concentration gradients can be explained

by the diurnal evolution of the planetary boundary layer. Usually gases accumulate at night when the planetary boundary layer is low due to cooler conditions, they then get diluted in greater volumes of the atmosphere within a well-mixed convective boundary layer that develops during daylight hours.

Diurnal variation in $CH_4$ show similar patterns to $CO_2$, with early morning maxima and early afternoon minima; however, the daytime increase in mole fraction with height as in Fig. 7(a)&(b) are not observed

in $CH_4$ (Fig. 7(c)&(d)) as mixing ratios are not dominated by biospheric photosynthesis. Average summer time $CH_4$ concentrations are ~ 20 nmol mol$^{-1}$ less than other seasons due to greater oxidation by hydroxyl radicals. In winter, vertical $CH_4$ concentration gradients are maintained throughout the day and night due the persistence of a low planetary boundary layer during daytime, caused by cooler atmospheric temperatures. Maximum gradients between the lowest and highest inlets show variation of

~ 10 and 20 nmol mol$^{-1}$ for RGL and TAC respectively, a similar percentage difference to $CO_2$ concentration gradients.





### 6.3 Discerning pollution events

Air history maps, showing the previous 30 days of surface influence at the station in a 1-hour period, were produced using the Met Office NAME Lagrangian atmospheric dispersion model (Jones et al., 2007) for each of the sites within the UK DECC network in order to discern and explain pollution

signals in the mole fraction measurements. Fig. 8(b) demonstrates a regionally polluted period at RGL for $CH_4$ on 30/11/2014, where air has passed over Europe and the south of the UK before arriving at the site. Fig. 8(c) shows an example of baseline conditions for $CH_4$ on 01/12/2014, where air passes over the North Atlantic Ocean, resulting in low mole fractions at MHD and variation between the other UK DECC sites.

### 7 Summary and conclusions

The UK DECC network was established in January 2012 to monitor atmospheric GHG and ODS mole fractions and verify the UK emission inventories submitted to the UNFCCC. The network was expanded from MHD, where GHG and ODS measurements have been made since 1987, to include RGL, Herefordshire, England; TAC, Norfolk, England; and TTA, Dundee, Scotland.

We have designed a network with robust systems for unattended continuous measurement of atmospheric $CO_2$, $CH_4$, $N_2O$, $SF_6$, CO and $H_2$ mixing ratios at tall open lattice telecommunications towers. Each of the UK DECC sites, excluding MHD, were equipped with CRDS instruments to measure $CO_2$ and $CH_4$ at ultra-high frequency (~ 3 seconds). RGL and TAC also had GC-ECD systems installed to measure $N_2O$ and $SF_6$ at high frequency (10 minutes). TAC and MHD also had a RGA

integrated with the GC-ECD system to measure CO and $H_2$. At MHD, a GC-ECD and –FID measure $N_2O$ and $CH_4$. At two selected sites (MHD and TAC), a Medusa GC-MS system was set up to measure $SF_6$ and a wide range of GHGs and ODSs. All instruments are controlled by Linux based software, GCWerks, and are accessible remotely.

Regular standard tank analysis shows CRDS precisions (1σ of 1-minute mean data of standard gas at

ambient mole fractions) of < 0.03 μmol mol$^{-1}$ and < 0.2 nmol mol$^{-1}$ for $CO_2$ and $CH_4$ respectively. GC-ECD instruments were also installed at RGL and TAC in January 2012 to measure $N_2O$ and $SF_6$.





Repeatability of 20 minute injections ($1\sigma$) between January 2012 and September 2015 for $N_2O$ and $SF_6$ measurements made using GC-ECD < 0.4 nmol mol$^{-1}$ and < 0.07 pmol mol$^{-1}$ respectively. Repeatability of 40/40/130 minute injections ($1\sigma$) between January 2012 and September 2015 for $CH_4$, $SF_6$, CO and $H_2$ measurements made using GC-FID/RGA/Medusa GC-MS were < 2.8 nmol mol$^{-1}$, < 2 nmol mol$^{-1}$

and <3 nmol mol$^{-1}$ and 0.34 pmol mol$^{-1}$, respectively.

Results from the network give good spatial and temporal coverage of atmospheric mixing ratios since January 2012. Results from the network show that all GHGs are increasing in concentration over the selected reporting period and, except for $SF_6$, exhibit seasonal trends. Discrete sample and hourly mean data are available from EBAS at http://ebas.nilu.no/ for GC-ECD/RGA and CRDS, respectively, for all

10 three UK sites. All MHD data, except CO mole fractions, are available from CDIAC (http://cdiac.ornl.gov/). The high-frequency data can be used for inversion modelling of GHG mole fractions and to discern GHG emissions for GHG inventory verification, as outlined in Manning et al. (2011).

## Acknowledgements

We specifically acknowledge the cooperation and efforts of the station operators Gerard Spain and Duncan Brown at Mace Head monitoring station, and Mr Stephen Humphrey and Mr Andy MacDonald at the Tacolneston tall tower station. We also thank the Physics Department, National University of Ireland, Galway, for making the research facilities at Mace Head available. The operation of all stations was funded by the UK Department of Business, Energy and Industrial Strategy (formerly the

Department of Energy and Climate Change) through contract TRN1028/06/2015 with additional funding at Mace Head under NASA contract NNX11AF17G through MIT with a sub award 5710002970 to the UoB.

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



Table 1: Greenhouse gas and ozone depleting substance species and instrumentation at each UK DECC site.

| Species | Sites | | | |
| --- | --- | --- | --- | --- |
| | Mace Head (MHD) | Tacolneston (TAC) | Ridge Hill (RGL) | Angus (TTA) |
| $CO_2$ | Picarro 2301[a] | Picarro G2301 | Picarro G2301 | Picarro G2301 |
| $CH_4$ | GC-FID[b] | Picarro G2301 | Picarro G2301 | Picarro G2301 |
| $N_2O$ | GC-ECD | GC-ECD | GC-ECD | - |
| $SF_6$ | Medusa | GC-ECD/Medusa | GC-ECD | - |
| $H_2$ | GC-RGA | GC-RGA | - | - |
| CO | GC-RGA | GC-RGA | - | - |
| $CF_4$ | Medusa | Medusa | - | - |
| $NF_3$ | Medusa | - | - | - |
| PFC-116 | Medusa | Medusa | - | - |
| PFC-218 | Medusa | Medusa | - | - |
| PFC-318 | Medusa | Medusa | - | - |
| HFC-23 | Medusa | Medusa | - | - |
| HFC-32 | Medusa | Medusa | - | - |
| HFC-134a | Medusa | Medusa | - | - |
| HFC-152a | Medusa | Medusa | - | - |
| HFC-125 | Medusa | Medusa | - | - |
| HFC-143a | Medusa | Medusa | - | - |
| HFC-227ea | Medusa | Medusa | - | - |
| HFC-236fa | Medusa | Medusa | - | - |
| HFC-43-10mee | Medusa | Medusa | - | - |
| HFC-365mfc | Medusa | Medusa | - | - |
| HCFC-22 | Medusa | Medusa | - | - |
| HCFC-141b | Medusa | Medusa | - | - |
| HCFC-142b | Medusa | Medusa | - | - |
| HCFC-124 | Medusa | Medusa | - | - |
| CFC-11 | Medusa | Medusa | - | - |
| CFC-12 | Medusa | Medusa | - | - |
| CFC-13 | Medusa | Medusa | - | - |
| CFC-113 | Medusa | Medusa | - | - |
| CFC-114 | Medusa | Medusa | - | - |
| CFC-115 | Medusa | Medusa | - | - |
| H-1211 | Medusa | Medusa | - | - |
| H-1301 | Medusa | Medusa | - | - |
| H-2402 | Medusa | Medusa | - | - |
| $CH_3Cl$ | Medusa | Medusa | - | - |
| $CH_3Br$ | Medusa | Medusa | - | - |
| $CH_3I$ | Medusa | Medusa | - | - |
| $CH_2Br_2$ | Medusa | Medusa | - | - |
| $CHCl_3$ | Medusa | Medusa | - | - |



| | | | | |
|---|---|---|---|---|
| CHBr$_3$ | Medusa | Medusa | - | - |
| CCl$_4$ | Medusa | Medusa | - | - |
| CH$_3$CCl$_3$ | Medusa | Medusa | - | - |
| CHCl=CCl$_2$ | Medusa | Medusa | - | - |
| CCl$_2$=CCl$_2$ | Medusa | Medusa | - | - |

[a] **Picarro G2301 instruments on site is owned and managed by Laboratoire des Sciences du Climat et de l'Environnement (LSCE), France. Data is available through the ICOS Carbon Portal (https://www.icos-cp.eu/).**

[b] **CH$_4$ is also analysed on the Picarro G2301 instrument maintained by LSCE. Data is available through the ICOS Carbon Portal (https://www.icos-cp.eu/).**



**Table 2: Site names, locations and inlet heights**

| Site Name | Acronym | Location | Altitude[a] (m.a.s.l) | Inlet Heights (m.a.g.l.) |
|---|---|---|---|---|
| Mace Head | MHD | 53.327 ° N 9.904 ° W | 8 | 10 |
| Ridge Hill tower | RGL | 51.998 ° N 2.540 ° W | 204 | 45, 90 |
| Tacolneston tower | TAC | 52.518 ° N 1.139 ° E | 56 | 54, 100, 185 |
| Angus tower | TTA | 56.555 ° N 2.986 ° W | 400 | 222 |

[a] **Altitude measured at base of tower.**





**Table 3: Gas chromatograph-flame ionisation and electron capture detector equipment and setup at UK DECC stations. * indicates $N_2O$ channel only on the MHD GC-ECD.**

|  | MHD | | TAC | RGL |
|---|---|---|---|---|
|  | FID | ECD* | ECD | ECD |
| Instrument | Carle AGC-211 | Hewlett-Packard 5890 | Agilent 6890N | Agilent 7890A |
| Detector | FID | ECD | μECD | μECD |
| Sample volume | 10 mL | 8 mL | 8 mL | 8 mL |
| Oven temperature | 60° C | 55° C | 90° C | 90° C |
| Column temperature | 60° C | 185° C (pre), 55° C (main) | 90° C (pre and main), 180° C (post) | 90° C (pre and main), 180° C (post) |
| Detector temperature | N/A | 325° C | 350° C | 350° C |
| Pre-column | Silica gel | Molecular sieve 5Å, 60/80 mesh | Porapack Q, 80/100 mesh, 1.0 m x 3/16" SS | Porapack Q, 80/100 mesh, 1.0 m x 3/16" SS |
| Main-column | Molecular sieve 5Å, 60/80 mesh | Porasil C | Porapack Q, 80/100 mesh, 2.0 m x 3/16" SS | Porapack Q, 80/100 mesh, 2.0 m x 3/16" SS |
| Post-column | N/A | N/A | Molecular sieve 5Å, 45/60 mesh, 0.9 m x 1/8" | Molecular sieve 5Å, 45/60 mesh, 0.9 m x 1/8" |
| Carrier gas supply | $N_2$ cylinder (5.0) | AR/$CH_4$ cylinder (95 %/5 %) (5.0) | AR/$CH_4$ cylinder (95 %/5 %) (5.0) | AR/$CH_4$ cylinder (95 %/5 %) (5.0) |
| $H_2$ supply | Cylinder (5.0) | N/A | N/A | N/A |
| Zero air supply | TOC generator (Parker Balston TOC-1250) | TOC generator (Parker Balston TOC-1250) | TOC generator (Parker Balston TOC-1250) | TOC generator (Parker Balston TOC-1250) |





**Table 4: Reduction gas analyser equipment and setup at UK DECC stations.**

|  | MHD | TAC |
| --- | --- | --- |
| Instrument | Trace Analytical RGA3 | Trace Analytical PP1 |
| Detector | RGA | RGA |
| Sample volume | 1 mL | 1 mL |
| Column temperature | 105˚ C | 105˚ C |
| Pre-column | Unibeads 1S, 60/80 mesh, 0.768 m x 1.8" | Unibeads 1S, 60/80 mesh, 0.768 m x 1.8" |
| Main-column | Molecular sieve 5Å, 60/80 mesh, 0.768 m x 1.8" | Molecular sieve 5Å, 60/80 mesh, 0.768 m x 1.8" |
| Carrier gas supply | Zero air cylinder (5.5) + purifier | Zero air cylinder (5.5) + purifier |
| Zero air supply | TOC generator (Parker Balston TOC-1250) | TOC generator (Parker Balston TOC-1250) |



Table 5: Cavity Ring-Down Spectrometry calibrant (cal) and standard gas mole fractions for $CO_2$ (µmol mol$^{-1}$) and $CH_4$ (nmol mol$^{-1}$) assigned by GASLAB at MPI-BGC Jena for the UK DECC network.

| Species | Gas | Site | | |
| --- | --- | --- | --- | --- |
| | | Tacolneston | Ridge Hill | Angus |
| | | (TAC) | (RGL) | (TTA) |
| | Standard | 386.70 | 385.44 | 401.29 |
| $CO_2$ | Cal 1 | 338.85 | 338.52 | 346.93 |
| (µmol mol$^{-1}$) | Cal 2 | 380.23 | 380.11 | 374.75 |
| WMO-X2007 | Cal 3 | 419.91 | 419.61 | 449.51 |
| | Cal 4 | 469.55 | 469.22 | - |
| | Standard | 1900.1 | 1953.7 | 1947.4 |
| $CH_4$ | Cal 1 | 1598.3 | 1598.2 | 1742.9 |
| (µmol mol$^{-1}$) | Cal 2 | 1797.3 | 17989.8 | 1851.5 |
| WMO-X2004A | Cal 3 | 1994.5 | 1992.0 | 2145.0 |
| | Cal 4 | 2189.2 | 2188.7 | - |



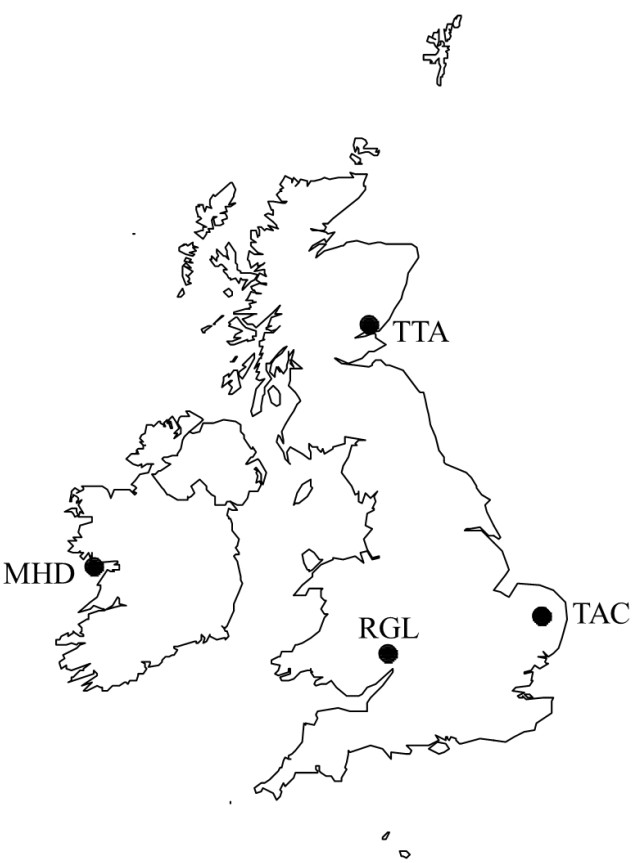

**Figure 1: Location of UK DECC network stations, showing from north to south: TTA, Angus, UK; MHD, Mace Head, Ireland; TAC, Tacolneston, UK; and RGL, Ridge Hill, UK.**





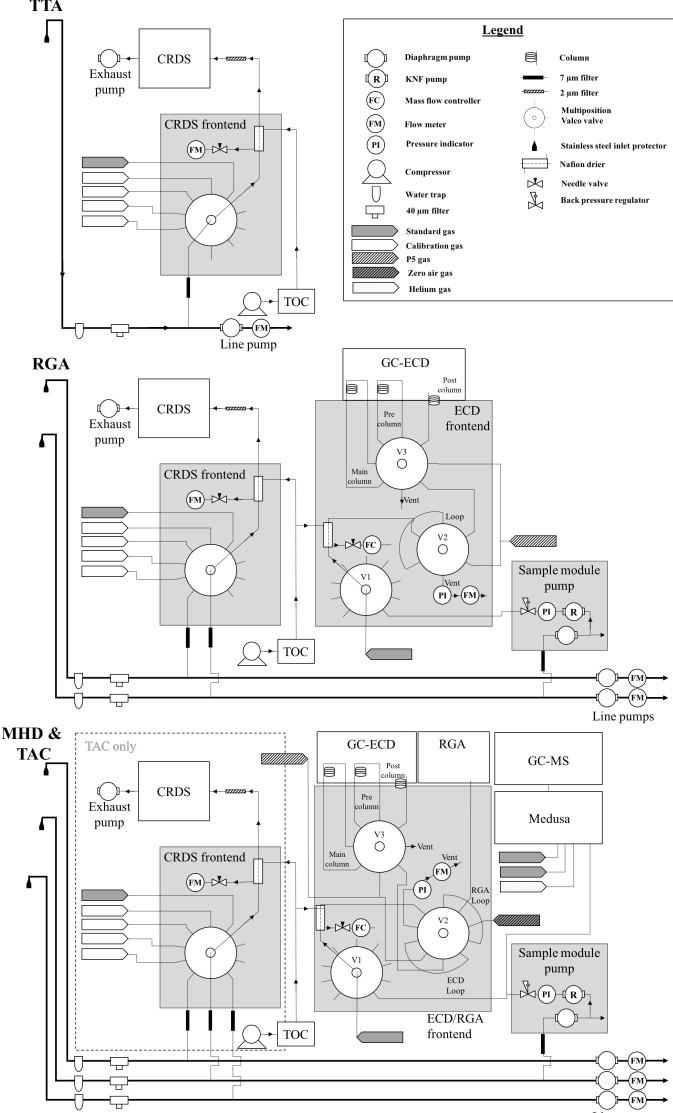

**Figure 2: Schematic diagram of the UK DECC network (Angus, TTA; Ridge Hill, RGL; and Tacolneston, TAC) CO$_2$, CH$_4$, N$_2$O, SF$_6$, CO and H$_2$ analysis system. The MHD setup is outlined in Prinn et al. (2000).**




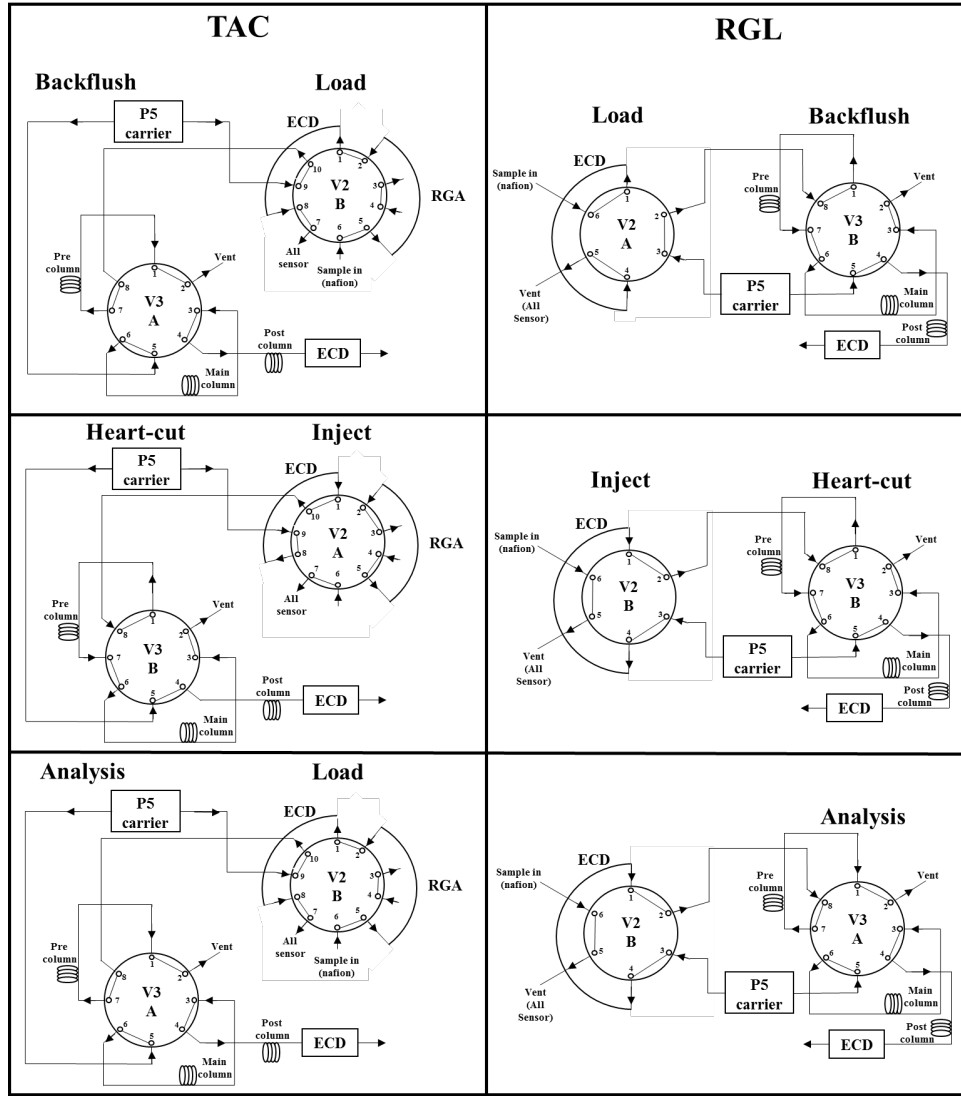

**Figure 3: ECD frontend valve configuration for sample backflush, heart-cut and analysis at Tacolneston (TAC) and Ridge Hill (RGL). The MHD setup is outlined in Prinn et al. (2000).**





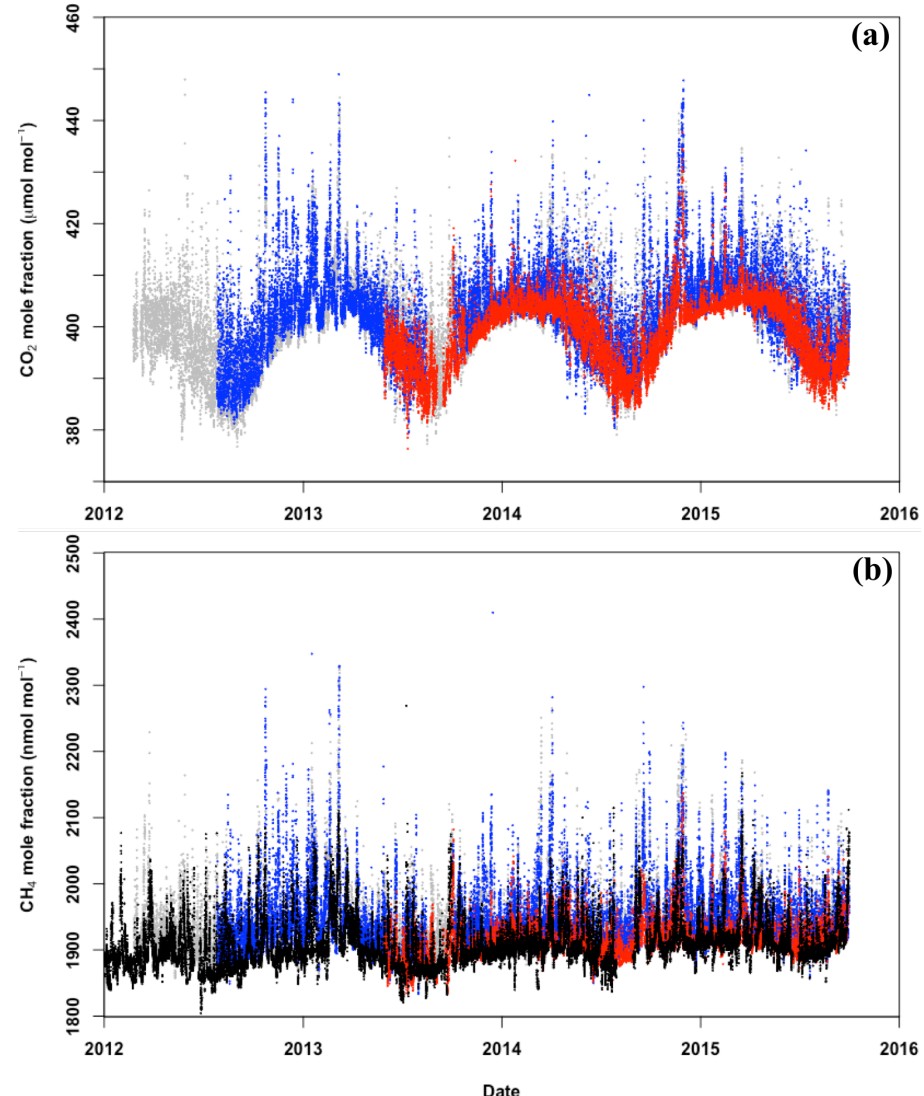

**Figure 4:** Time series of (a) $CO_2$ and (b) $CH_4$ mole fractions at MHD (black; 10 m inlet) RGL (grey; 90 m inlet), TTA (red; 222 m inlet) and TAC (blue; 100 m inlet) from 1 January 2012 to 30 September 2015. Results shown are hourly averages. $CO_2$ results shown are in µmol mol$^{-1}$ and are on the WMO-X2007 scale. $CH_4$ results are shown in nmol mol$^{-1}$ and are on the WMO-X2004A scale.





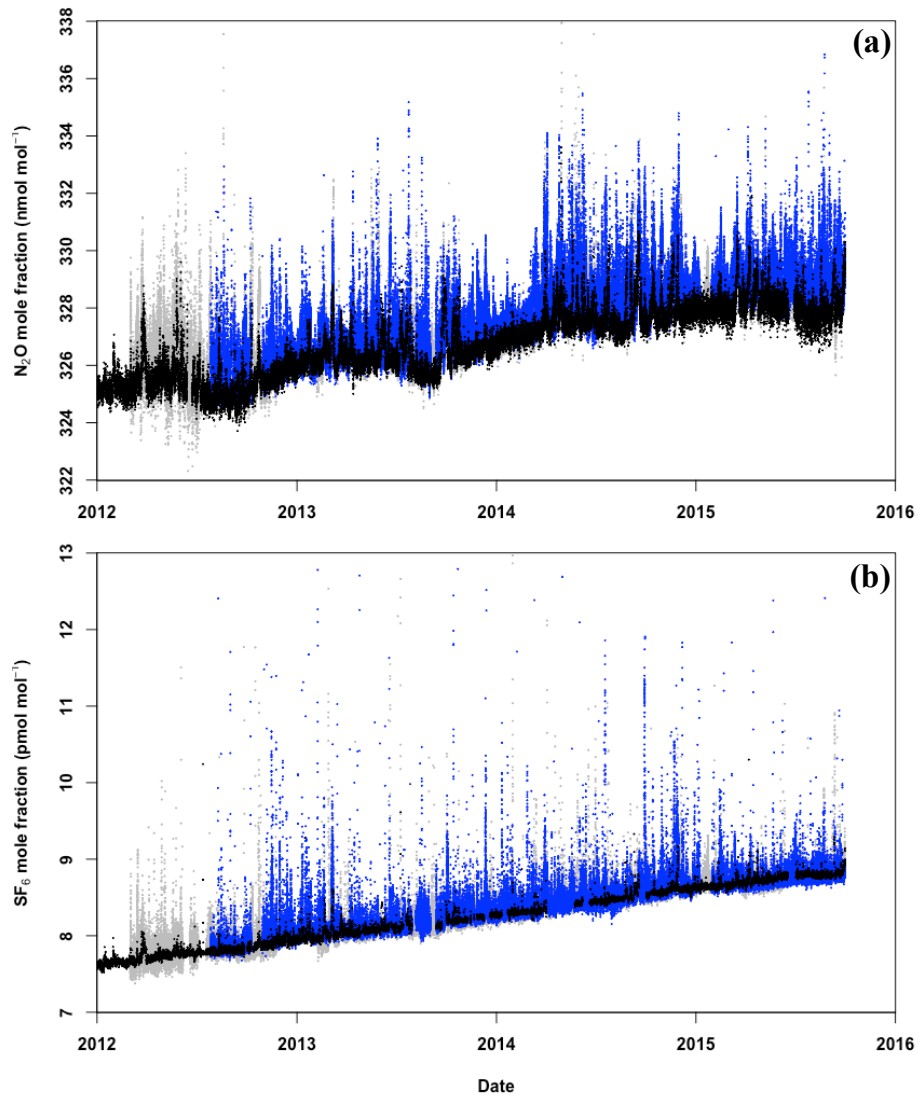

**Figure 5: Time series of (a) N₂O and (b) SF₆ mole fractions at MHD (black; 10 m inlet), RGL (grey; 90 m inlet) and TAC (blue; 100 m inlet) from 1 January 2012 to 30 September 2015. All results shown are hourly averages. N₂O results are shown in nmol mol⁻¹ and are on the SIO-98 scale, whilst SF₆ results are shown in pmol mol⁻¹ and are on SIO-05 scale. SF₆ results from MHD are made using the Medusa GC-MS.**



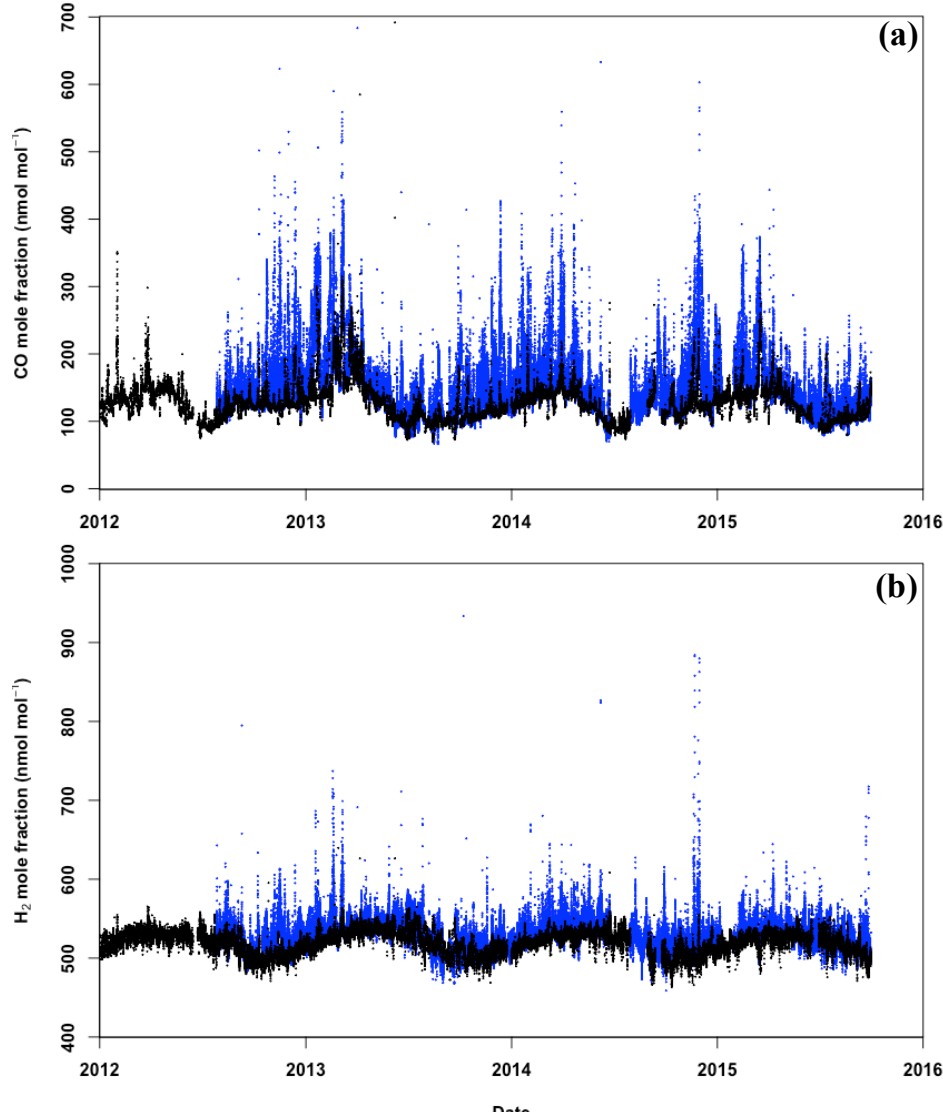

**Figure 6: Time series of (a) CO and (b) H₂ mole fractions at MHD (black; 10 m inlet) and TAC (blue; 100 m inlet) from 1 January 2012 to 30 September 2015. All results shown are hourly averages, shown in nmol mol⁻¹ and are on CSIRO04 and MPI-2009 scales for CO and H₂, respectively.**





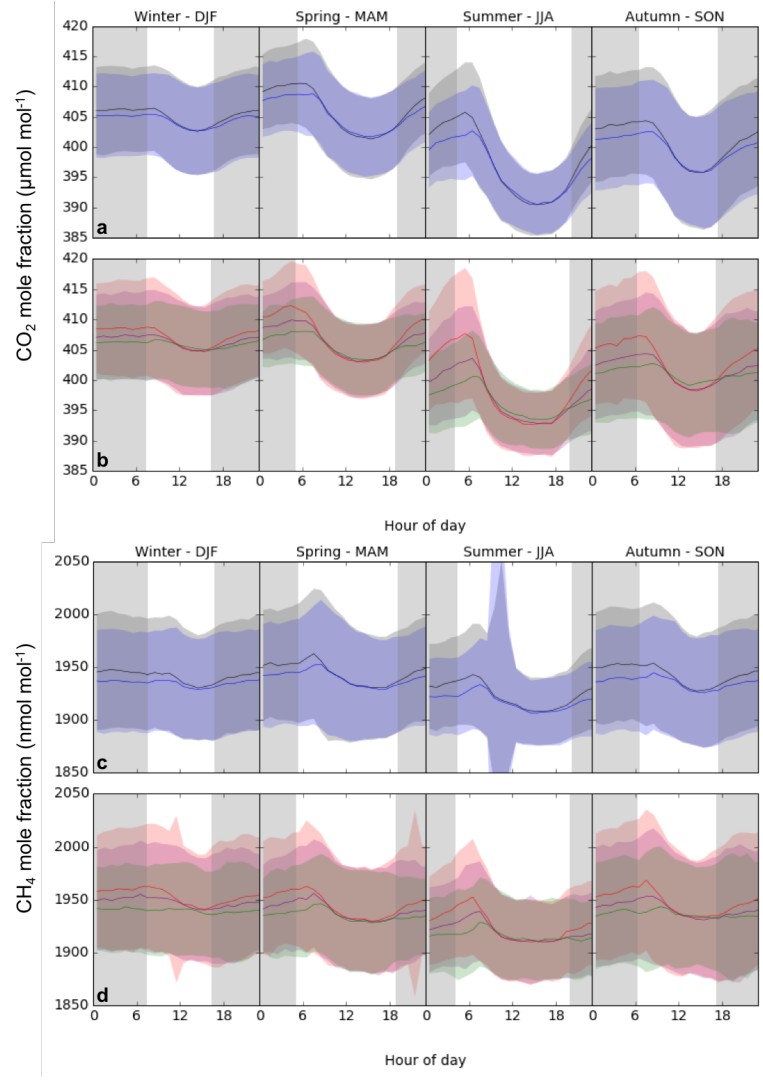

**Figure 7: Overview of average diurnal concentration gradients in CO$_2$ at (a) RGL and (b) TAC, and CH$_4$ at (c) RGL and (d) TAC from 23 January 2012 to 1 October 2015. Lines are the median of the entire data period and shaded areas represent the standard deviation (1 σ). Black and blue data correspond to the 45 and 90 m.a.g.l inlet at RGL, respectively, and red, purple and green correspond to the 54, 100 and 185 m.a.g.l inlets at TAC. Shaded grey areas represent mean seasonal night-time based on the sites latitude and longitude.**





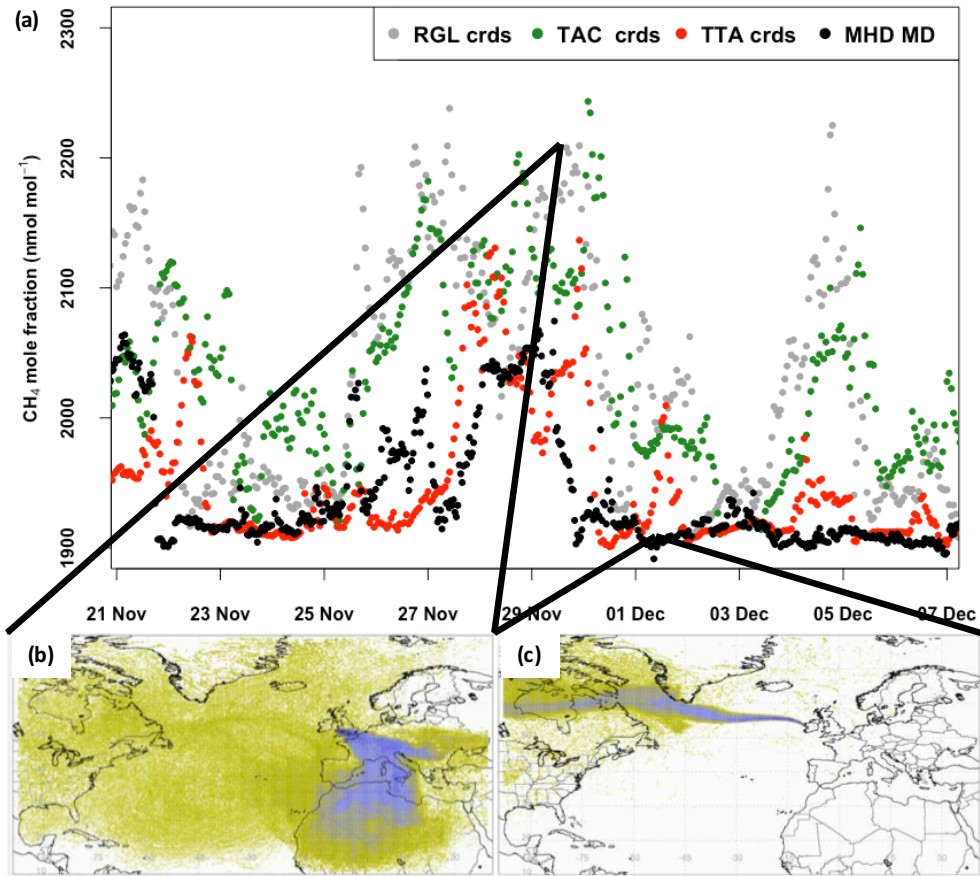

**Figure 8:** Examples of (a) CH$_4$ mole fractions from 21$^{st}$ November to 7$^{th}$ December 2014 (MHD MD discrete samples, CRDS values 20 minute averages), and 2-hour air history maps derived from NAME (b) for RGL, regionally polluted period, and (c) for MHD, baseline period. The air-history maps describe which surface areas (0-40m) in the previous 30-days impact the observation point within a particular 2-hour period.