# Peer review of "Greenhouse gas measurements from a UK network of tall towers: technical description and first results"

_Atmospheric Measurement Techniques, 2017_

## Referee Comment (RC1) · Anonymous Referee #1 · 18 Nov 2017

review of amt-2017-349

Title: Greenhouse gas measurements from a UK network of tall towers: technical description and first results

Authors: Stanley et al.

General comments:

The manuscript presents a comprehensive overview of the recently set-up network of greenhouse gas measurements at tall towers in the UK. The paper is clearly structured and well written. It provides much useful information for readers that also aim

at establishing monitoring capabilities at tall towers. Therefore, the manuscript merits publication in Atmospheric Measurement Techniques. However, the paper summarizes several things already published in the literature. Thus, I feel that some paragraphs of the manuscript could be shortened by minimizing repetition of already published technical description.

At the same time, the paper somehow lacks novelty or the author missed to emphasize the new approaches applied here. I suggest to highlight the new approaches and, in addition, to provide more information likely of interest for the reader. Issues to be addressed in this respect are (a) the required maintenance of the measurements (e.g. how did the regular maintenance look like, how many maintenance (regular as well as "emergency") visits were needed over the years); (b) how did the troubleshooting look like; (c) which major technical problems were the authors facing; and (d) add a paragraph on lessons learnt in the conclusions.

How does this exercise refer to European efforts like ICOS, in terms of instrumentation, quality control and data processing?

Moreover, there are also a few minor comments that should be considered prior to publication.

Specific comments:

Page 1, line 17: I don't agree with the wording "automated custom-built instrumentation" here as a large part of the instruments is commercially available.

Page 2, lines 11-13: "...independent emission estimates for comparison with the UK national inventory ...": if this is mentioned in the abstract, I expected to see some related results in the manuscript but I couldn't find it.

Page 4, line 6: "...except MHD which samples every 20 minutes ...": this statement is in contradiction to Table 1 where a Picarro G2301 is listed for MHD.

Page 5, line 19 – 20: Here for RGL and also further below for TAC: what is the rationale

for measuring at several heights?

Page 7, lines 3 – 10: why Synflex tubing is used at RGL and TAC while stainless steel tubing is used at MHD. What are the pros and cons for using one of them? What is the difference between Synflex 1300 and Synflex 3000?

Page 7, line 17: I suppose that only the bowl is made out of Perspex. What is the material used for the other wetted parts like the body, the seals and the gaskets?

Page 9, line 22 to page 10, line 1: If the temperature is maintained at 318 +/- 0.004 K, line 1 must read a few / thousandth of a K ..., correct?

Page 10, line 11: 0.25 % volume ratio of water: which dew point is that?

Page 10, lines 12 – 14: did you test for potential CO2 losses in the Nafion?

Page 10, lines 17 – 21: Correction for water vapour interferences: are the correction coefficients listed in Rella (2010) the ones that are implemented in the Picarro software? If so, the direct Picarro output internally corrected for H2O cross-talk can be used right away, correct? Did you also use the Rella (2010) factors for TTA when running the measurements w/o dryer? If so, why didn't you use individually determined correction factors as suggested by Rella et al. (2013) (https://www.atmos-meas-tech.net/6/837/2013/).

Page 12, lines 11 – 12: It reads like it was a new idea of the authors to use 5% CH4 in Ar (i.e. P5) instead of CO2-doped N2 as carrier gas. However, this is already done by many groups for many years. You may refer to Schmidt et al. (2001) (JGR; http://onlinelibrary.wiley.com/doi/10.1029/2000JD900701/epdf).

Page 16, line 13: what is a "significant change"? Was the change detected automatically or manually (visually)? Which criterion was applied?

Page 16, lines 14 – 15: "A second order non-linear curve is fit to the data ...": Does that mean that the Picarro has a non-linear response? If so, please state it clearly

and elaborate on it and quantify the effect when erroneously neglecting the non-linear response.

Page 17, lines 9 – 11: If I understand correctly, only a one point calibration approach is applied. This only works when assuming no detector signal at zero concentration, right. Was that tested and how was the one point calibration approach applied when a GC-MS signal > 0 was detected for species-free air?

Page 17, lines 25 – 26: how did you make sure that there were no traces of N2O and SF6 in the zero air? The zero air, was it real air having N2O and SF6 trapped or was it a N2/O2 mixture? How about the Argon content in the zero air?

Page 18, 8: troubleshooting is mentioned here but it remains too vague if and how and how often troubleshooting was required. See one of my general comments above.

Page 18, line 18: I suggest skipping the explanation of the naming convention of the stripchart files. This is irrelevant.

Page 18, line 23: remove "with time stamps corresponding to the beginning of the measurement, and stored" It isn't of importance here.

Page 19, line 5: add a table with the thresholds for the maximum allowed standard deviations?

Page 19, line 24: remove explanation of the naming convention.

Page 20, line 1: is it important that there was a 4:1 compression ratio?

Page 21, lines 3 – 5: how often did it happen that the processing routines filtered false negatives? Is that a time consuming (and important) task to review the automatically filtered data?

Page 21, lines 9 – 10: Is the flagging of spurious data a manual process? If so, I suggest clarifying it by saying "Spurious data are manually flagged and a justification . . . can be added and logged."

Page 21, line 11: what is "GCcompare"?

Page 22, lines 18 – 19: where are the 21 nomol mol-1 coming from? Add a reference.

Page 23, lines 2 -3: Add a statement that the trend of 0.8 nmol mol-1 per year is lower than the global trend. You may refer to the latest WMO GHG bulletin #12 (https://library.wmo.int/opac/doc_num.php?explnum_id=3084). What do you mean by seasonal trend, isn't it the annual growth rate?

Page 23, chapter 6.2: this is all largely text-book knowledge and can be considerably shortened.

Page 25, Summary and conclusions: It only summarizes what was said before. I would like to read some kind of outlook and some recommendations that go beyond a simple description of the setup as given above. Topics to be potentially addressed could be: are there any modifications planned (based on some lessons –learnt); are there any major flaws in the setup which cannot be easily changed anymore; with the experience gained during the few years of operation, would the setup again look the same when you may be able to once more start from scratch?

Page 26, lines 11 – 13: The reference given here to underline the benefit for such measurements for GHG inventory verification was published in 2011, i.e. before the presented measurements were implemented. Either remove the reference to the emission verification on page 26 (and in the abstract) or elaborate on the benefit of additional observations for the GHG inventory assessment based on tall-tower measurements and inverse modelling.

Page 31, footnote a to Table 1: I cannot find the data on the ICOS carbon portal as indicated.

---

## Referee Comment (RC2) · Anonymous Referee #2 · 22 Dec 2017

Review of the manuscript: "Greenhouse gas measurements from a UK network of tall towers: technical description and first results", submitted to Atmos. Meas. Tech., by Kieran M. Stanley et al.

The paper is describing the setup for greenhouse gases measurements at four sites in UK. The authors are providing a very detailed description of the inlet parts, analyzers, and calibration protocols. In the last section of the paper atmospheric signals (diurnal and seasonal cycles, trend) are very briefly discussed. Considering the purpose of the manuscript I am not fully convinced by the need for this very general discussion on the observed variabilities, coming after very detailed technical descriptions of the setup

and protocols. From my point of view, the most problematic point of this manuscript is the lack of analysis of the measurements in terms of uncertainties and quality control. There are very few quantitative indicators which could help to justify some choices in the protocols. For CRDS measurements only precisions estimates are provided, whereas repeatability's are provided for GCs. Even more problematic is the use of the precision (based on standard deviations calculated over one minute intervals) as a 'demonstration' that measurements are compliant with WMO recommendation for compatibility of monitoring sites. But it seems that few efforts were put to characterize the possible biases (e.g. no measurement of a target gas as recommended by WMO/GAW). I would expect at least some analysis of the existing information (e.g. variability of the calibration residuals), description of the troubleshooting, etc... Also I am very surprised some suspicious signals are not discussed at all, even though the protocol described in the manuscript claims a significant emphasis of the data review.

Page 1, line 14: "A network of ...three... tall tower"

Page 1, line 24: "The long-term 1-minute mean precisions (1s)" You should show an indicator comparable to GC

Page 3, line 2: "the accuracy of the inversion is limited by the number and distribution of measurement locations available": I would suggest to mention also the capacity of models to properly represent the observed time series, which lead to favor tall towers over flat terrains.

Page 3, line 9: "Measurements...constrained global or hemispheric scale fluxes": I would not say that atmospheric measurements constrained the fluxes (also on the next line of the same paragraph). Hopefully they can give some constraints to the estimation of the fluxes by inverse method.

Page 4, line 3: I do not think the list of species measured by Medusa is needed in Table 1. Please refer to a publication.

[Figure]

Page 5, line 4: 51% of marine air at MHD: please give a reference

Page 6, line 2: Is there a reason why you chose to sample N2O, SF6, CO at 100m a.g.l. and not at the highest point of the tower (185m). Also, why did not you adopt the same sapling strategy at TTA where you have a single inlet for CO2, CH4, whereas you set up 2 or inlets at the other tall towers ? Could you please explain your choices ?

Page 7, line 10: What do you mean by: 'Horizontal sections of tubing at the base of the tower were kept to a minimum' ? Please clarify.

Page 7, line 17: Have you tested the Perspex H2O decanting bowls to ensure the non-contamination of the measurements ? Why did you changed from the previous system used at TTA ?

Figure 2: Can you precise the meaning of TOC on the figures.

Page 8, line 20: "This has the advantage of eliminating sample contamination from the pump, reducing the likelihood of a torn diaphragm introducing laboratory air into the sample and improving the performance of Nafion dryers." Can you please elaborate on this sentence ? The first point is clear, but I understand that the inlet is slightly under pressure which is not an advantage to avoid the contamination from lab air.

Page 10, line 20: "The correction applied is minimized due to the removal of most H2O using the Nafion dryer." In return you introduced a possible CO2 bias due to the nafion. How have you quantified this bias ?

Page 12, line 17: "Each cylinder is now individually analysed as a sample to check for contamination prior to use". Can you precise how frequently you get problem with the purity of the carrier gas ?

Page 15: "4.1 Sampling sequence" : Is there any justification of the different configurations at the sites. For example why 20 or 30 min sampling time ? Is it because of the number of sampling levels ? Also could you clarify if the GCs are measuring at only one level ?

Page 16, line 5: Can you precise the lifetimes of the standard and calibration gases like you have done for the GCs standards ? Are the standards mixing ratios reevaluated at the end ?

Page 16, line 9: "line flushing": are you choices in duration, number of cycle and frequency of the calibration gases measurements based on specific tests ?

Page 16, line 16: "long-term precision": what about the measurement repeatability ?

Page 16, line 19: "CRDS precision within the UK DECC network is within the WMO compatibility guidelines for CO2 ($\pm$ 0.1 $\mu$mol mol-1) and CH4 ($\pm$ 2 nmol mol-1) (WMO-GAW, 2014)": Indeed the precision is lower than the WMO recommendation for compatibility, but this is comparing apples and oranges. The WMO compatibility has to take into account the measurement repeatability, non-linearity, calibration uncertainties, H2O correction, dryer and inlet biases. Please remove or rephrase correctly this sentence.

Regarding WMO recommendations it should be noted that you are not following the recommendation of measuring a target tank ("Each analysis system must include at least one 'target tank' which is a very important quality control tool ", WMO/GAW report n° 229, 2015)

Page 17, line 16: "Calibration scales vary depending on the gas species": the CH4 scale for GC-FID measurements is not given. Is it a different scale compared to CRDS measurements ? If so what about the compatibility of the two scales ? Also the CH4 is not mentioned at all in paragraph 5.2 (GC data processing).

Page 17, line 24: "periodically in the field": Can you precise the frequency of the non-linearity tests for N2O and SF6 ?

Page 19, line 4: "H2O level too high": what is the reason of rejecting values with high H2O level ? what is the typical threshold values ?

Page 19, line 14: "A second order function can then be fitted to the data to provide

a non-linearity correction": CRDS instruments have the reputation to be quite linear. Why do you apply a non-linear correction? Have you estimated the importance of this term ?

Page 20, line 11: "using sample integrated height or area": can you precise which one you are using for the different species ?

Page 23, line 3: "There is an approximate 0.8 nmol mol-1 northern hemisphere mid-latitude seasonal trend": if you give an estimate of the trend for N2O, I would suggest to do the same for all species, and gather this information in a table.

Figure 5: How do you explain the low N2O and SF6 concentrations measured at RGL in the first year (2012) ? Those data look suspicious considering that such event do not appear later on. According to your description of the final data processing (5.3) you are evaluating the time series by comparing the stations with MHD. From my understanding this is typically a case where such comparison could lead to flagging the first year of measurements. Can you provide a possible explanation of this atmospheric signal ?

Figure 7: Please precise if you are using local or UTC time on this figure.

Figure 7c: CH4 diurnal cycle at RGL shows a very high standard deviation in mid-day/summer. Outliers which could cause such anomaly on the 2012/2015 average signal are not visible in the time series on figure 4.

Page 25, line 5: "Fig. 8(b) demonstrates a regionally polluted period at RGL for CH4 on 30/11/2014, where air has passed over Europe and the south of the UK before arriving at the site": in such a case we could expect higher CH4 concentrations at MHD compared to TAC, due to the additional contribution from south UK. Do you such 'reverse' gradients sometimes ?

Page 26, line 6: "Results from the network give good spatial and temporal coverage": I fully agree for the temporal resolution, but I would remove the comment on the spatial coverage, the relevance of which depends on the intended scientific purpose.

---

## Author Comment (AC1) · 18 Jan 2018

**Reply to the comments of Reviewer #1 on the manuscript "Greenhouse gas measurements from a UK network of tall towers: technical description and first results"**

Kieran M. Stanley[1,] Aoife Grant[1], Simon O'Doherty[1], Dickon Young[1], Alistair J. Manning[1,2], Ann R. Stavert[1], T. Gerard Spain[3], Peter K. Salameh[4], Christina M. Harth[4], Peter G. Simmonds[1], William T. Sturges[5], David E. Oram[5], Richard G. Derwent[6]

[1]School of Chemistry, University of Bristol, Bristol, United Kingdom
[2]Met Office, Exeter, Devon, United Kingdom
[3]Department of Experimental Physics, National University of Ireland, Galway, Ireland
[4]Scripps Institution of Oceanography, University of California San Diego, La Jolla, California, USA
[5]School of Environmental Sciences, University of East Anglia, Norwich, United Kingdom
[6]rdscientific, Newbury, Berkshire, United Kingdom

We thank the reviewer #1 for their time and effort in evaluating this manuscript and for their suggestions for improvements. All points made by the reviewer are addressed on the following pages.

**The manuscript presents a comprehensive overview of the recently set-up network of greenhouse gas measurements at tall towers in the UK. The paper is clearly structured and well written. It provides much useful information for readers that also aim at establishing monitoring capabilities at tall towers. Therefore, the manuscript merits publication in Atmospheric Measurement Techniques. However, the paper summarizes several things already published in the literature. Thus, I feel that some paragraphs of the manuscript could be shortened by minimizing repetition of already published technical description.**

Response: It is understood that the paper summarises a number of things already published in the literature and that some paragraphs could be shortened.

Modification: A number of sections have been modified, as outlined below:

- Page 7, line 4: "This tubing is made from high-density polyethylene bonded to overlapped aluminium tape and has a total wall thickness of 1.57 mm (Andrews et al., 2014). The outer polyethylene coating makes it resistant to water ($H_2O$) vapour condensation on the inner aluminium core tube." was removed.
- Page 7, line 13: "For each inlet at RGL and TAC, an inverted stainless steel cup covers the inlet, acting as a shield to prevent $H_2O$ entering the line. A monel mesh screen is inserted within the cup to help prevent large particles from entering the inlet lines." was changed to "For each inlet at RGL and TAC, an inverted stainless steel cup with a monel mesh screen inserted within the cup covers the inlet, acting as a shield to prevent $H_2O$ entering the line."
- Page 7 line 23-24: "The filters are present to protect instruments and pumps from particles (Andrews et al., 2014)" was removed.
- Page 10, line 3-10: "$H_2O$ can damage system components and interfere with measurements of GHGs, even at low levels (Andrews et al., 2014). $H_2O$ influences

mole fractions of GHGs measured via CRDS through a dilution effect, whereby a difference of 100 μmol mol$^{-1}$ (or parts per million; ppm) $H_2O$ can cause a "dilution offset" of 0.04 μmol mol$^{-1}$ in $CO_2$ (Andrews et al., 2014). Also, $H_2O$ vapour differences between calibration gases and air samples can cause spectral issues within optical instruments. $H_2O$ causes these spectral artefacts through line broadening effects on the spectroscopic line shapes, specifically Lorentzian broadening and Dicke line narrowing (Chen et al., 2010; Rella et al., 2013). The extent of these effects is dependent on the atmospheric mixing ratio of $H_2O$ (Chen et al., 2010)" changed to "$H_2O$ can damage system components and interfere with measurements of GHGs, even at low levels, through a dilution effect (Andrews et al., 2014) and pressure broadening effects (Chen et al., 2010; Rella et al., 2013)".

- Page 10, line 13: "The effectiveness of permeation Nafion dryers is dependent on the $H_2O$ partial pressure gradient between the sample and counter purge flows (Andrews et al., 2014)" was removed.

- Page 11, line 7 – page 12, line 10: two paragraphs changed to "The $SF_6$ and $N_2O$ analysis method used at RGL and TAC was similar to that described in detail in Ganesan et al. (2013), except P-5 carrier gas (a mixture of 5 % $CH_4$ in 95 % Ar; Air Products, UK) is used. Briefly, calibration gas and air samples are flushed through an 8 mL sample loop at 40 mL min$^{-1}$ for 60 seconds at a fixed exhaust pressure (~ 20 psi; Fig. 3 Backflush) before decaying down to ambient pressure. Flow through the loop is controlled by a 'RED-y' smart series mass flow controller (GSC-A4TA-BB22, Voeglin Instruments AG, Switzerland) and pressure in the loop is measured using an 'All Sensor' pressure sensor (100PSI-A-DO, All Sensors, BS-Rep GmbH, Germany). Once equilibrated to ambient pressure, samples are then injected through an 8-port, 2-position valve (V3 in Fig. 2; EUDAC8UWEPH, VICI Valco AG International, Switzerland) onto a pre- (1.0 m Porapak Q, 80/100 mesh, 3/16" O.D.) and main column (2.0 m Porapak Q, 80/100 mesh, 3/16" O.D.) held at 90 °C, where $N_2O$ and $SF_6$ are separated from air. Oxygen is "heart-cut" to vent (V3, Fig. 3) after it has eluted from the two columns, whilst the pre-column is back flushed with P5. The remaining, $O_2$ minimised, sample flows through the post-column (0.9 m of 1/8" O.D. stainless steel packed with molecular sieve 5Å, 45/60 mesh), housed in a thermostatically controlled heated inlet port of the GC at 180 °C. The post column reverses the elution order of $SF_6$ and $N_2O$ to prevent the larger $N_2O$ peak from tailing into the small $SF_6$ peak, improving sample reproducibility and precision. Detection occurs in the ECD which is held at 350 °C. The ECDs at TAC and RGL measure at a rate of 10 and 20 Hz respectively. Samples are dried using a Nafion Dryer (MD-050-72S-1, Perma Pure, USA) with a dry zero air counter purge (as outlined in Sect. 3.3)".

- Page 12, line 12: "The use of P5 carrier gas enables the omission of $CO_2$ doping" was removed.

- Page 12, line 13: "$N_2O$ co-elutes with $CO_2$ on the column combination used within the UK DECC network, saturating the MS 5Å and providing a constant doping effect, thus care is taken to make sure the $N_2O$ response if not affected by this" was changed to "$N_2O$ co-elutes with $CO_2$ on the post-column, saturating the MS 5Å and providing a constant doping effect and reducing precision".

- Page 12, line 23 – page 13, line 11: paragraph changed to "CO and $H_2$ are measured at two sites, MHD and TAC, using a RGA (RGA3 (MHD) and Peak Performer 1 (TAC), Trace Analytical Inc., USA). Table 4 outlines RGA instrumental setup at TAC and MHD. The MHD RGA setup is different to TAC and is outlined in (Prinn et al., 2000). The TAC sample selection system is integrated within the GC-ECD

system (Sect. 3.4) (Grant et al., 2010a; Grant et al., 2010b). The GC-ECD has a 10-port, 2-position valve (VICI Valco AG International, Switzerland) for V2 (Fig. 2), instead of an 8-port 2-position valve, as at RGL. This allows for a 1 mL RGA sample loop to be put in sequence before the ECD sample loop (Fig. 3 TAC). After samples have been dried using the Nafion Dryer (MD-050-72S-1, Perma Pure, USA), passed through the sample loops and decayed to ambient pressure, they are injected onto two isothermal packed columns held at 105 °C: a 0.768 m pre-column (1/8" O.D. stainless steel packed with 60/80 mesh Unibeads 1S) and a 0.768 m main column (1/8" O.D. stainless steel packed with MS 5Å, 60/80 mesh). After separation, gases are injected into the RGA for analysis using zero air plus (Air Products, UK) carrier gas, where the samples pass over a heated bed of mercuric oxide before being quantitatively determined using UV photometry (Grant et al., 2010a; Grant et al., 2010b)".

**At the same time, the paper somehow lacks novelty or the author missed to emphasize the new approaches applied here. I suggest to highlight the new approaches and, in addition, to provide more information likely of interest for the reader. Issues to be addressed in this respect are (a) the required maintenance of the measurements (e.g. how did the regular maintenance look like, how many maintenance (regular as well as "emergency") visits were needed over the years); (b) how did the troubleshooting look like; (c) which major technical problems were the authors facing; and (d) add a paragraph on lessons learnt in the conclusions**

Response: Two extra sections have been added to include information on maintenance (section 3.8; Table 5) and troubleshooting data issues (section 5.4; Table 8) as suggested.

**How does this exercise refer to European efforts like ICOS, in terms of instrumentation, quality control and data processing?**

Response: Instrumentation used within the UK DECC network is very similar to those used for $CO_2$ and $CH_4$ measurements in ICOS (Yver Kwok et al., 2015; Hazan et al., 2016) and other non-European measurement programmes, such as the Los Angeles Mega City Project (Verhulst et al., 2017). However, the data processing and quality control is different within ICOS to the UK DECC network. The ICOS method calibrates data by linear interpolation between a linear fit of a suite of calibration gases (n = 4) spanning above and below ambient concentrations (Hazan et al., 2016), whereas the method used in the UK DECC network uses a daily standard to calibrate out daily instrumental drift and then suite of calibration cylinders to correct for effects above and below ambient mole fractions.

**Specific comments:**
**Page 1, line 17: I don't agree with the wording "automated custom-built instrumentation" here as a large part of the instruments is commercially available.**

Response: "custom-built" removed.

**Page 2, lines 11-13: ". . .independent emission estimates for comparison with the UK national inventory . . .": if this is mentioned in the abstract, I expected to see some related results in the manuscript but I couldn't find it.**

Response: This sentence has been removed as these modelling results are outside of the scope of this paper.

**Page 4, line 6: ". . .except MHD which samples every 20 minutes . . .": this statement is in contradiction to Table 1 where a Picarro G2301 is listed for MHD.**
     Response: "Except MHD which samples every 20 minutes" has been removed.

**Page 5, line 19 – 20: Here for RGL and also further below for TAC: what is the rationale for measuring at several heights?**
     Response: $CO_2$ and $CH_4$ are measured at several heights on the tall towers to try and asses for boundary layer stratification. Measurements at several inlet heights with instruments that are able to measure at high-frequency has the added benefit of aiding troubleshooting of data.

**Page 7, lines 3 – 10: why Synflex tubing is used at RGL and TAC while stainless steel tubing is used at MHD. What are the pros and cons for using one of them? What is the difference between Synflex 1300 and Synflex 3000?**
     Response: Historically, stainless steel (SS) has been used at MHD when it was set up as a sampling site for AGAGE in 1994, as well as for the predecessor research programmes, as SS does not outgas any of the halocarbons measured in the research programme. The relatively short run of tubing from the inlet to the line pump at MHD meant that ¼" i.d. SS tubing gave the desired flow rates with the line pumps used. However, for the tall towers, a wider i.d. tubing (½" i.d.) was necessary to get the necessary flow rates to flush the tubing on the tall towers well enough. The weight of ½" i.d. stainless steel tubing on the tall towers, as well as the impracticality and financial cost of installing a continuous length of ½" i.d. SS tubing, Synflex tubing was used instead. Double bonded aluminium core tubing, such as Synflex, is used routinely on tall tower sites (it is recommended in the ICOS specifications: http://icos-atc.lsce.ipsl.fr/filebrowser/download/27251) due to its flexibility, light weight and cost.
     The difference between Synflex 1300 and Synflex 3000 is the material used for the inner core of the tubing. The former has a polyethylene inner core, whilst the latter has a nylon inner core. Both plastics outgas; hence, the flow rates at the sites were kept high (~ 20 L min$^{-1}$) to ensure that no build-up of trace gases being measured at the sites occurred.

**Page 7, line 17: I suppose that only the bowl is made out of Perspex. What is the material used for the other wetted parts like the body, the seals and the gaskets?**
     Response: Correct, only the bowl is made of Perspex on the $H_2O$ decanting bowl. The other wetted parts of the system are made of the following: the housing is aluminium, the filter is sintered polypropylene and the seals are either nitrile or neoprene. The sentence on page 7, line 17, has been reworded to make it clearer what the wetted parts of the system are.

**Page 9, line 22 to page 10, line 1: If the temperature is maintained at 318 +/- 0.004 K, line 1 must read a few / thousandth of a K . . ., correct?**
     Response: Correct, changed to thousandths.

**Page 10, line 11: 0.25 % volume ratio of water: which dew point is that?**
     Response: 0.25 % $H_2O$ is -10.3˚C dew point. This has been added to the sentence in parentheses.

**Page 10, lines 12 – 14: did you test for potential CO2 losses in the Nafion?**

Response: No tests were undertaken by the authors themselves but we had spoken to collaborators at AGAGE and Scripps Institute of Oceanography about the use of Nafion dryers and their tests into cross membrane leakage of $CO_2$ and $CH_4$; work presented in Welp et al. (2013). $CO_2$ and $CH_4$ bias from cross-membrane permeation was observed by the authors; however, the procedure of passing both the sample air and calibration gases through the Nafion cancels out the bias (Welp et al., 2013).

**Page 10, lines 17 – 21: Correction for water vapour interferences: are the correction coefficients listed in Rella (2010) the ones that are implemented in the Picarro software? If so, the direct Picarro output internally corrected for H2O cross-talk can be used right away, correct? Did you also use the Rella (2010) factors for TTA when running the measurements w/o dryer? If so, why didn't you use individually determined correction factors as suggested by Rella et al. (2013) (https://www.atmos-meastech.net/6/837/2013/).**

Response: The correction coefficients listed in (Rella, 2010) are implemented in the Picarro software to derive $CO_2$ and $CH_4$ dry mole fractions and were used at TTA to correct data to dry mole fractions. No individually determined correction factor were assessed or implemented at TTA due to time constraints and site access issues.

**Page 12, lines 11 – 12: It reads like it was a new idea of the authors to use 5% CH4 in Ar (i.e. P5) instead of CO2-doped N2 as carrier gas. However, this is already done by many groups for many years. You may refer to Schmidt et al. (2001) (JGR; http://onlinelibrary.wiley.com/doi/10.1029/2000JD900701/epdf).**

Response: The sentence was reworded as part of the first comment on minimising repetition of previously published work. The above reference has also been added to the sentence so that it now reads: "The $SF_6$ and $N_2O$ analysis method used at RGL and TAC was similar to that described in detail in Ganesan et al. (2013), except P-5 carrier gas (a mixture of 5 % $CH_4$ in 95 % Ar; Air Products, UK) is used (Schmidt et al., 2001)."

**Page 16, line 13: what is a "significant change"? Was the change detected automatically or manually (visually)? Which criterion was applied?**

Response: Any changes in the coefficients from the curve of the nonlinearity correction would be classed as a significant change. The word significant was removed from the sentence to make the sentence clearer. Changes were detected manually when comparing the previous coefficients with the new coefficients. The sentence was altered to make this clearer.

**Page 16, lines 14 – 15: "A second order non-linear curve is fit to the data . . .": Does that mean that the Picarro has a non-linear response? If so, please state it clearly and elaborate on it and quantify the effect when erroneously neglecting the non-linear response.**

Response: For all of the UK DECC network sites there is a small non-linear response on the CRDSs. The non-linear response is not thought to be due to calibration gases used within the network as the non-linear effect has been seen when NOAA standards have been used to calibrate instruments that use GCWerks. An extra figure has been added to shown an

example of the non-linear fits used within the network (Figure 4) and histograms to show the offsets between data with and without the non-linear corrections (Figure 5). Additional information has been put in the paragraph to exemplify what the effect would be on the data if the non-linear correction wasn't applied. There was a small difference between the two steps in data correction, with a median value of $< -0.01$ μmol mol$^{-1}$ $CO_2$ and $< -0.002$ nmol mol$^{-1}$ $CH_4$ (linear corrected - non-linear corrected data).

**Page 17, lines 9 – 11: If I understand correctly, only a one point calibration approach is applied. This only works when assuming no detector signal at zero concentration, right. Was that tested and how was the one point calibration approach applied when a GC-MS signal > 0 was detected for species-free air?**

      Response: Yes, the one point calibration approach does assume that no detector signal at zero concentration for the Medusa GC-MS and was tested when the Medusa system was being developed at Scripps Institute of Oceanography (Miller et al., 2008; Arnold et al., 2012). Weekly, a system blank is analysed on the Medusa GC-MS, whereby He is passed through the pre-concentration system instead of a sample gas to the GC-MS. The system blank allows the detection of signals >0 in species-free gas and GCWerks can then integrate the contamination peak. A blank correction can then be implemented to cancel out the contamination.

**Page 17, lines 25 – 26: how did you make sure that there were no traces of N2O and SF6 in the zero air? The zero air, was it real air having N2O and SF6 trapped or was it a N2/O2 mixture? How about the Argon content in the zero air?**

      Response: The zero air used for the non-linearity tests was analysed for traces of $N_2O$ and $SF_6$ during the non-linearity test sequence. Any cylinders with detectable $N_2O$ or $SF_6$ contamination were not used. A sentence has been added to clarify this. The zero air used for the tests was a synthetic blend of $N_2/O_2$, with Ar contents $< 0.01$ μmol mol$^{-1}$.

**Page 18, 8: troubleshooting is mentioned here but it remains too vague if and how and how often troubleshooting was required. See one of my general comments above.**

      Response: Created section 5.4 to discuss troubleshooting of data within the network.

**Page 18, line 18: I suggest skipping the explanation of the naming convention of the stripchart files. This is irrelevant.**

      Response: Removed.

**Page 18, line 23: remove "with time stamps corresponding to the beginning of the measurement, and stored" It isn't of importance here.**

      Response: Removed

**Page 19, line 5: add a table with the thresholds for the maximum allowed standard deviations?**

      Response: Added in Table 7 with parameter thresholds for all CRDS filters.

**Page 19, line 24: remove explanation of the naming convention.**
Response: Removed

**Page 20, line 1: is it important that there was a 4:1 compression ratio?**
Response: Removed

**Page 21, lines 3 – 5: how often did it happen that the processing routines filtered false negatives? Is that a time consuming (and important) task to review the automatically filtered data?**
Response: When the data filters were introduced to the software and the parameterisation of the filters had not been finely tuned, there were a number of occasions when the air filter was not set high enough and real pollution events were filtered out. Through reviewing the stripcharts, this was spotted and the standard deviation filter was increased to 10. It is important to check all of the data, including filtered data, to ensure that no instrumental problems are missed due to the filtering of data. Within GCWerks, all data is shown within the stripcharts, so the viewing of the stripcharts is not that time consuming (a day's worth of data can usually be checked in about 5 minutes).

**Page 21, lines 9 – 10: Is the flagging of spurious data a manual process? If so, I suggest clarifying it by saying "Spurious data are manually flagged and a justification . . . can be added and logged."**
Response: "manually" added.

**Page 21, line 11: what is "GCcompare"?**
Response: GCcompare is data visualisation software, where data within a tab delimited format can be imported to the visualisation software and the time series data can be overlain with other time series data from other sites for the same compound to look and investigate errors.

**Page 22, lines 18 – 19: where are the 21 nmol mol-1 coming from? Add a reference.**
Response: The amplitude data is from Mace Head. This has been clarified in the sentence.

**Page 23, lines 2 -3: Add a statement that the trend of 0.8 nmol mol-1 per year is lower than the global trend. You may refer to the latest WMO GHG bulletin #12 (https://library.wmo.int/opac/doc_num.php?explnum_id=3084). What do you mean by seasonal trend, isn't it the annual growth rate?**
Response: There is a 0.8 nmol mol$^{-1}$ amplitude in the seasonal trend at MHD, not the annual growth rate. The sentence has been altered to clarify this.

**Page 23, chapter 6.2: this is all largely text-book knowledge and can be considerably shortened.**

Response: The section has been shortened.

**Page 25, Summary and conclusions: It only summarizes what was said before. I would like to read some kind of outlook and some recommendations that go beyond a simple description of the setup as given above. Topics to be potentially addressed could be: are there any modifications planned (based on some lessons –learnt); are there any major flaws in the setup which cannot be easily changed anymore; with the experience gained during the few years of operation, would the setup again look the same when you may be able to once more start from scratch?**

Response: A new recommendations section has been added (section 7) to discuss recommendations based on experiences within the network. Additionally, future work planned in the network has been added the conclusion.

**Page 26, lines 11 – 13: The reference given here to underline the benefit for such measurements for GHG inventory verification was published in 2011, i.e. before the presented measurements were implemented. Either remove the reference to the emission verification on page 26 (and in the abstract) or elaborate on the benefit of additional observations for the GHG inventory assessment based on tall-tower measurements and inverse modelling.**

Response: Removed

**Page 31, footnote a to Table 1: I cannot find the data on the ICOS carbon portal as indicated.**

Response: When accessed on 12/01/18, data was found to be available in the ICOS data portal search (https://data.icos-cp.eu/portal/#search?station=%5B%22Mace%20Head%20%22%5D).

[revised manuscript text omitted]

Welp, L. R., Keeling, R. F., Weiss, R. F., Paplawsky, W., and Heckman, S.: Design and performance of a Nafion dryer for continuous operation at $CO_2$ and $CH_4$ air monitoring sites, Atmos. Meas. Tech., 6, 1217-1226, 10.5194/amt-6-1217-2013, 2013.

Yver Kwok, C., Laurent, O., Guemri, A., Philippon, C., Wastine, B., Rella, C. W., Vuillemin, C., Truong, F., Delmotte, M., Kazan, V., Darding, M., Lebègue, B., Kaiser, C., Xueref-Rémy, I., and Ramonet, M.: Comprehensive laboratory and field testing of cavity ring-down spectroscopy analyzers measuring $H_2O$, $CO_2$, $CH_4$ and CO, Atmos. Meas. Tech., 8, 3867-3892, 10.5194/amt-8-3867-2015, 2015.

---

## Author Comment (AC2) · 18 Jan 2018

**Reply to the comments of Reviewer #2 on the manuscript "Greenhouse gas measurements from a UK network of tall towers: technical description and first results"**

Kieran M. Stanley[1,] Aoife Grant[1], Simon O'Doherty[1], Dickon Young[1], Alistair J. Manning[1,2], Ann R. Stavert[1], T. Gerard Spain[3], Peter K. Salameh[4], Christina M. Harth[4], Peter G. Simmonds[1], William T. Sturges[5], David E. Oram[5], Richard G. Derwent[6]

[1]School of Chemistry, University of Bristol, Bristol, United Kingdom
[2]Met Office, Exeter, Devon, United Kingdom
[3]Department of Experimental Physics, National University of Ireland, Galway, Ireland
[4]Scripps Institution of Oceanography, University of California San Diego, La Jolla, California, USA
[5]School of Environmental Sciences, University of East Anglia, Norwich, United Kingdom
[6]rdscientific, Newbury, Berkshire, United Kingdom

We thank the reviewer #2 for their time and effort in evaluating this manuscript and for their suggestions for improvements. All points made by the reviewer are addressed on the following pages.

**The paper is describing the setup for greenhouse gases measurements at four sites in UK. The authors are providing a very detailed description of the inlet parts, analyzers, and calibration protocols. In the last section of the paper atmospheric signals (diurnal and seasonal cycles, trend) are very briefly discussed. Considering the purpose of the manuscript I am not fully convinced by the need for this very general discussion on the observed variabilities, coming after very detailed technical descriptions of the setup and protocols. From my point of view, the most problematic point of this manuscript is the lack of analysis of the measurements in terms of uncertainties and quality control. There are very few quantitative indicators which could help to justify some choices in the protocols. For CRDS measurements only precisions estimates are provided, whereas repeatability's are provided for GCs. Even more problematic is the use of the precision (based on standard deviations calculated over one minute intervals) as a 'demonstration' that measurements are compliant with WMO recommendation for compatibility of monitoring sites. But it seems that few efforts were put to characterize the possible biases (e.g. no measurement of a target gas as recommended by WMO/GAW). I would expect at least some analysis of the existing information (e.g. variability of the calibration residuals), description of the troubleshooting, etc. . . Also I am very surprised some suspicious signals are not discussed at all, even though the protocol described in the manuscript claims a significant emphasis of the data review.**

   Response: The aim of this paper was to predominantly describe the setup of the UK DECC network, including the instrumentation used, sampling and calibration protocols and data processing methods to help guide the setup of other future sites and networks. The addition of the data since the network starts help show what data there is available and the potential uses for it. No in depth analysis was presented in this paper of the data as the

authors felt that this would be better done in a paper where interesting signals and patterns can be looked at in conjunction with inverse modelling results.

No quantification of uncertainty within the data has been presented in this paper as this is something that the authors want to work on to include with the data. This has been outlined in the final paragraph of the summary and conclusions, where we have added in the future improvements for the network. We realise the added benefit of using target tanks as an independent quality control measure; however, when the network was set up, we used protocols based on GC measurements within AGAGE, which has not historically measured target tanks. The inclusion of target tanks within the network is another aspect that we are looking into to include within the network.

We recognise the issues related to the use of precision information for CRDS measurements and repeatability values for GC measurements. We have changed our precision data to short-term precision and included repeatability data for the CRDS instruments, which is outlined in the specific comments below. In addition, we have removed the comparison with the WMO recommendations as we realise that our precision data does not fully encapsulate the error from measurements in the network.

Three extra sections have been added to this manuscript to improve information on the reader, including a troubleshooting section (section 5.4) and a section on maintenance (section 3.8) and a recommendations section for readers wishing to create future sites or stations.

**Page 1, line 14: "A network of . . .three. . . tall tower"**
Response: Done.

**Page 1, line 24: "The long-term 1-minute mean precisions (1s)" You should show an indicator comparable to GC**
Response: This has been changed to short-term precision ($1\sigma$ of 1 minute mean data per standard injection) and a repeatability of standard injections ($1\sigma$ of 20-minute mean injections).

**Page 3, line 2: "the accuracy of the inversion is limited by the number and distribution of measurement locations available": I would suggest to mention also the capacity of models to properly represent the observed time series, which lead to favor tall towers over flat terrains.**
Response: Done.

**Page 3, line 9: "Measurements. . .constrained global or hemispheric scale fluxes": I would not say that atmospheric measurements constrained the fluxes (also on the next line of the same paragraph). Hopefully they can give some constraints to the estimation of the fluxes by inverse method.**
Response: Text altered to give clarity for both sentences.

**Page 4, line 3: I do not think the list of species measured by Medusa is needed in Table 1. Please refer to a publication.**
Response: Medusa species removed, apart from $SF_6$, and Miller et al. (2008) and Arnold et al. (2012) papers were referred to.

**Page 5, line 4: 51% of marine air at MHD: please give a reference**
  Response: Reference added

**Page 6, line 2: Is there a reason why you chose to sample N2O, SF6, CO at 100m a.g.l. and not at the highest point of the tower (185m). Also, why did not you adopt the same sapling strategy at TTA where you have a single inlet for CO2, CH4, whereas you set up 2 or inlets at the other tall towers ? Could you please explain your choices ?**
  Response: The reason for sampling $N_2O$, $SF_6$ and CO at the 100 m.a.g.l. inlet is purely historical. When the site was setup in 2012, the 185 m.a.g.l. inlet was not installed. A sentence has been added about the later installation of the 185m inlet. Multiple inlets were chosen at sites except TTA to try and assess boundary layer stratification and is a protocol used by a number of different networks with tall tower sites, including ICOS. The added advantage of having multiple inlets is that if a line pump fails or a filter blocks on an inlet, the affected line can be taken out of the sampling strategy. Additionally, data from other inlets can be used to help diagnose issues in data by looking at patterns between each inlet (see section 5.4 for more details). The sampling line set up at TTA was different as the site was inherited by the University of Bristol and the cost of adding additional sampling lines was too much for that site.

**Page 7, line 10: What do you mean by: 'Horizontal sections of tubing at the base of the tower were kept to a minimum' ? Please clarify.**
  Response: From the base of the towers, the tubing runs horizontally along trunking until it reaches the laboratory. We tried to minimise these long horizontal stretches to prevent water from accumulating and the potential for contaminants to alter the composition of the air drawn down from the inlet.

**Page 7, line 17: Have you tested the Perspex H2O decanting bowls to ensure the noncontamination of the measurements ? Why did you changed from the previous system used at TTA ?**
  Response: Yes, the Perspex $H_2O$ decanting bowls were tested in the laboratory before being installed at. A steady gas stream of ambient air (whole air compressed into a cylinder) was passed through Synflex 1300 tubing and analysed on a CRDS with and without the $H_2O$ decanting bowls in line to observe for differences in $CO_2$ and $CH_4$ mole fractions. No effect was found. A sentence has been added to state that the bowls were tested before being installed. Perspex $H_2O$ decanting bowls were used in preference over the stainless steel bowls to enable site operators to see if $H_2O$ had accumulated within the bowl.

**Figure 2: Can you precise the meaning of TOC on the figures.**
  Response: TOC (zero air generator) has been added to the figure legend.

**Page 8, line 20: "This has the advantage of eliminating sample contamination from the pump, reducing the likelihood of a torn diaphragm introducing laboratory air into the sample and improving the performance of Nafion dryers." Can you please elaborate on this sentence ? The first point is clear, but I understand that the inlet is slightly under pressure which is not an advantage to avoid the contamination from lab air.**

Response: The Nafion section was removed as it was found to be incorrect.

**Page 10, line 20: "The correction applied is minimized due to the removal of most H2O using the Nafion dryer." In return you introduced a possible CO2 bias due to the nafion. How have you quantified this bias ?**

Response: The passing of calibration and standard gases through the Nafion dryer makes bias associated with $CO_2$ permeation across the membrane negligible (Welp et al., 2013; Andrews et al., 2014). Laboratory tests by Andrews et al. (2014) showed that losses of $CO_2$ from sample air and calibration gases were the same.

**Page 12, line 17: "Each cylinder is now individually analysed as a sample to check for contamination prior to use". Can you precise how frequently you get problem with the purity of the carrier gas ?**

Response: The frequency of $SF_6$ contamination within P5 varies greatly as it depends on the cylinder age. $SF_6$ was used to pressure test cylinders nearing the end of their inspection date to check for leaks and cylinder stability. On average, about one in every six cylinders were contaminated with trace amounts of $SF_6$. A section was also added in the recommendations about testing carrier gases before use (section 7.4).

**Page 15: "4.1 Sampling sequence" : Is there any justification of the different configurations at the sites. For example why 20 or 30 min sampling time ? Is it because of the number of sampling levels ? Also could you clarify if the GCs are measuring at only one level ?**

Response: As stated within the text, the justification for different configurations in CRDS sampling times is to ensure that all inlet heights are measured within one hour and is purely based on the number of sampling heights at the site. A sentence has been added to clarify that the GC are only sampled from one height.

**Page 16, line 5: Can you precise the lifetimes of the standard and calibration gases like you have done for the GCs standards ? Are the standards mixing ratios reevaluated at the end ?**

Response: A sentence has been added to the previous paragraph (first paragraph of section 4.2.1) stating the lifetimes of CRDS calibration and standard gases. A sentence has also been added stating that cylinders are recalibrated once removed from site.

**Page 16, line 9: "line flushing": are you choices in duration, number of cycle and frequency of the calibration gases measurements based on specific tests ?**

Response: Choices were made according to suggestions made by ICOS when setting up the network rather than specific tests. Specifications used by ICOS are available in Hazan et al. (2016).

**Page 16, line 16: "long-term precision": what about the measurement repeatability ?**

Response: This has been changed to long-term repeatability based on standard injections on the CRDS. A short-term precision has also been included.

**Page 16, line 19: "CRDS precision within the UK DECC network is within the WMO compatibility guidelines for CO2 (± 0.1 µmol mol-1) and CH4 (± 2 nmol mol-1) (WMOGAW, 2014)": Indeed the precision is lower than the WMO recommendation for compatibility, but this is comparing apples and oranges. The WMO compatibility has to take into account the measurement repeatability, non-linearity, calibration uncertainties, H2O correction, dryer and inlet biases. Please remove or rephrase correctly this sentence. Regarding WMO recommendations it should be noted that you are not following the recommendation of measuring a target tank ("Each analysis system must include at least one 'target tank' which is a very important quality control tool ", WMO/GAW report n ∘ 229, 2015)**

Response: Sentence removed.

**Page 17, line 16: "Calibration scales vary depending on the gas species": the CH4 scale for GC-FID measurements is not given. Is it a different scale compared to CRDS measurements ? If so what about the compatibility of the two scales ? Also the CH4 is not mentioned at all in paragraph 5.2 (GC data processing).**

Response: $CH_4$ measured on the MHD GC-FID is on the Tohoku University scale. Based on in situ measurements made at MHD by AGAGE and NOAA flask measurements since 1993, there is an average difference of 1.01 nmol mol$^{-1}$ (NOAA-AGAGE) (Krummel, 2018). $CH_4$ has been added into section 5.2.

**Page 17, line 24: "periodically in the field": Can you precise the frequency of the nonlinearity tests for N2O and SF6 ?**

Response: Tests were conducted approximately every year. The sentence has been altered to reflect this.

**Page 19, line 4: "H2O level too high": what is the reason of rejecting values with high H2O level ? what is the typical threshold values ?**

Response: Data was filtered out when $H_2O$ values were > 6% and was rejected as this was the highest mixing ratio used in the $H_2O$ correction in Rella (2010) . Additionally, data with such a high $H_2O$ concentration was indicative of liquid water passing the $H_2O$ decanting bowl. Table 7 was added to show filter parameters.

**Page 19, line 14: "A second order function can then be fitted to the data to provide a non-linearity correction": CRDS instruments have the reputation to be quite linear. Why do you apply a non-linear correction? Have you estimated the importance of this term ?**

Response: For all of the UK DECC network sites there is a small non-linear response on the CRDSs. The non-linear response is not thought to be due to calibration gases used within the network as the non-linear effect has been seen when NOAA standards have been

used to calibrate instruments that use GCWerks. An extra figure has been added to shown an example of the non-linear fits used within the network (Figure 4) and histograms to show the offsets between data with and without the non-linear corrections (Figure 5). Additional information has been put in the paragraph to exemplify what the effect would be on the data if the non-linear correction wasn't applied.

**Page 20, line 11: "using sample integrated height or area": can you precise which one you are using for the different species ?**
      Response: Done

**Page 23, line 3: "There is an approximate 0.8 nmol mol-1 northern hemisphere midlatitude seasonal trend": if you give an estimate of the trend for N2O, I would suggest to do the same for all species, and gather this information in a table.**
      Response: This sentence has been altered to show the amplitude of the seasonal cycle.

**Figure 5: How do you explain the low N2O and SF6 concentrations measured at RGL in the first year (2012) ? Those data look suspicious considering that such event do not appear later on. According to your description of the final data processing (5.3) you are evaluating the time series by comparing the stations with MHD. From my understanding this is typically a case where such comparison could lead to flagging the first year of measurements. Can you provide a possible explanation of this atmospheric signal ?**
      Response: Within the UK DECC network, data is not manually flagged out unless there is a specific reason for the spurious data, such as instrumental issues or leaks, even if the data looks odd against MHD. Instead, if no reason can be found, the data is left in the time series. The $N_2O$ and $SF_6$ data shown in Figure 5 (now Figure 7) from the start of the time series to 11/07/2012 is noisier due to poorer instrument precision, which improved in July 2012 when more insulation was added to around the inlet of the post-column. Hence why both gases have periods with lower mole fractions than MHD.

**Figure 7: Please precise if you are using local or UTC time on this figure.**
      Response: Done, UTC.

**Figure 7c: CH4 diurnal cycle at RGL shows a very high standard deviation in midday/summer. Outliers which could cause such anomaly on the 2012/2015 average signal are not visible in the time series on figure 4.**
      Response: Plot altered to show all of data.

**Page 25, line 5: "Fig. 8(b) demonstrates a regionally polluted period at RGL for CH4 on 30/11/2014, where air has passed over Europe and the south of the UK before arriving at the site": in such a case we could expect higher CH4 concentrations at MHD compared to TAC, due to the additional contribution from south UK. Do you such 'reverse' gradients sometimes ?**

Response: If both sites are receiving the same air masses, then an increasing gradient of $CH_4$ mole fractions would be expected between RGL and MHD. However, in the instance shown in Figure 8(b), now figure 10(b), the air mass that RGL received was from over the midlands and south of the UK and the Benelux region and Germany; however, at this time, MHD was received air that had passed over the south west of the UK, the west of France and Spain as shown in Figure 1 below. The reverse gradient from that shown in Figure 10(b) – an increasing concentration from west to east over the UK, is frequently observed due to the UK predominant wind direction. Figure 10(a) demonstrates the increasing $CH_4$ mole fraction with longitude on the 4[th] and 5[th] December. A sentence has been added to section 6.3 to reflect this.

[Figure]

**Figure 1: 2-hour air history maps derived from NAME for MHD on 29[th] November 2014. The air-history maps describe which surface areas (0-40m) in the previous 30-days impact the observation point within a particular 2-hour period.**

**Page 26, line 6: "Results from the network give good spatial and temporal coverage": I fully agree for the temporal resolution, but I would remove the comment on the spatial coverage, the relevance of which depends on the intended scientific purpose.**
Response: Spatial has been removed.

**References**
Andrews, A. E., Kofler, J. D., Trudeau, M. E., Williams, J. C., Neff, D. H., Masarie, K. A., Chao, D. Y., Kitzis, D. R., Novelli, P. C., Zhao, C. L., Dlugokencky, E. J., Lang, P. M., Crotwell, M. J., Fischer, M. L., Parker, M. J., Lee, J. T., Baumann, D. D., Desai, A. R., Stanier, C. O., De Wekker, S. F. J., Wolfe, D. E., Munger, J. W., and Tans, P. P.: CO2, CO, and CH4 measurements from tall towers in the NOAA Earth System Research Laboratory's Global Greenhouse Gas Reference Network: instrumentation, uncertainty analysis, and recommendations for future high-accuracy greenhouse gas monitoring efforts, Atmos. Meas. Tech., 7, 647-687, 10.5194/amt-7-647-2014, 2014.
Arnold, T., Mühle, J., Salameh, P. K., Harth, C. M., Ivy, D. J., and Weiss, R. F.: Automated Measurement of Nitrogen Trifluoride in Ambient Air, Analytical Chemistry, 84, 4798-4804, 10.1021/ac300373e, 2012.

Hazan, L., Tarniewicz, J., Ramonet, M., Laurent, O., and Abbaris, A.: Automatic processing of atmospheric CO2 and CH4 mole fractions at the ICOS Atmospheric Thematic Center, Atmos. Meas. Tech. , 9, 1-34, 10.5194/amt-9-4719-2016, 2016.

Krummel, P.: RE: MHD NOAA-AGAGE difference for CH4. Personal communication to Stanley, K. M. on 15/01/2018.

Miller, B. R., Weiss, R. F., Salameh, P. K., Tanhua, T., Greally, B. R., Muhle, J., and Simmonds, P. G.: Medusa: A Sample Preconcentration and GC/MS Detector System for in Situ Measurements of Atmospheric Trace Halocarbons, Hydrocarbons, and Sulfur Compounds, Anal. Chem., 80, 1536-1545, 2008.

Rella, C.: Accurate greenhouse gas measurements in humid gas streams using the Picarro G1301 carbon dioxide/methane/water vapor gas analyzer, White paper, Picarro Inc, Sunnyvale, CA, USA, 2010.

Welp, L. R., Keeling, R. F., Weiss, R. F., Paplawsky, W., and Heckman, S.: Design and performance of a Nafion dryer for continuous operation at CO2 and CH4 air monitoring sites, Atmos. Meas. Tech., 6, 1217-1226, 10.5194/amt-6-1217-2013, 2013.

---

## Author Response (AR2)

**Reply to the comments of Anonymous Referee #1's report on the manuscript "Greenhouse gas measurements from a UK network of tall towers: technical description and first results"**

Kieran M. Stanley[1*,] Aoife Grant[1], Simon O'Doherty[1], Dickon Young[1], Alistair J. Manning[1,2], Ann R. Stavert[1], T. Gerard Spain[3], Peter K. Salameh[4], Christina M. Harth[4], Peter G. Simmonds[1], William T. Sturges[5], David E. Oram[5], Richard G. Derwent[6]

[1]School of Chemistry, University of Bristol, Bristol, United Kingdom
[2]Met Office, Exeter, Devon, United Kingdom
[3]Department of Experimental Physics, National University of Ireland, Galway, Ireland
[4]Scripps Institution of Oceanography, University of California San Diego, La Jolla, California, USA
[5]School of Environmental Sciences, University of East Anglia, Norwich, United Kingdom
[6]rdscientific, Newbury, Berkshire, United Kingdom

We thank the Referee for their technical corrections and suggestions on this manuscript. The 0.6 % from section 4.2.2. has been removed so that page 17, lines 6-9 now read: "A concentration difference of 1.01 ± 4.14 
[revised manuscript text omitted]